# Discovery of a molecular glue promoting CDK12-DDB1 interaction to trigger cyclin K degradation

Lu Lv[1,2†], Peihao Chen[2,3†], Longzhi Cao[2,4†], Yamei Li[2,4], Zhi Zeng[2,4], Yue Cui[1,2], Qingcui Wu[2], Jiaojiao Li[2], Jian-Hua Wang[2], Meng-Qiu Dong[2,5], Xiangbing Qi[2,5*], Ting Han[2,4,5*]

[1]College of Life Sciences, Beijing Normal University, Beijing, China; [2]National Institute of Biological Sciences, Beijing, China; [3]School of Life Sciences, Peking University, Beijing, China; [4]Graduate School of Peking Union Medical College and Chinese Academy of Medical Sciences, Beijing, China; [5]Tsinghua Institute of Multidisciplinary Biomedical Research, Tsinghua University, Beijing, China

**Abstract** Molecular-glue degraders mediate interactions between target proteins and components of the ubiquitin-proteasome system to cause selective protein degradation. Here, we report a new molecular glue HQ461 discovered by high-throughput screening. Using loss-of-function and gain-of-function genetic screening in human cancer cells followed by biochemical reconstitution, we show that HQ461 acts by promoting an interaction between CDK12 and DDB1-CUL4-RBX1 E3 ubiquitin ligase, leading to polyubiquitination and degradation of CDK12-interacting protein Cyclin K (CCNK). Degradation of CCNK mediated by HQ461 compromised CDK12 function, leading to reduced phosphorylation of a CDK12 substrate, downregulation of DNA damage response genes, and cell death. Structure-activity relationship analysis of HQ461 revealed the importance of a 5-methylthiazol-2-amine pharmacophore and resulted in an HQ461 derivate with improved potency. Our studies reveal a new molecular glue that recruits its target protein directly to DDB1 to bypass the requirement of a substrate-specific receptor, presenting a new strategy for targeted protein degradation.

**\*For correspondence:**
qixiangbing@nibs.ac.cn (XQ);
hanting@nibs.ac.cn (TH)

[†]These authors contributed equally to this work

## Introduction

Molecular glues are a class of small molecules that induce the formation of protein-protein interactions to elicit biologic or therapeutic effects. The name 'molecular glue' was first coined to describe the mechanism of action of the plant hormone auxin, which bridges an interaction between the E3 ubiquitin ligase TIR1 and IAA transcription repressors, leading to IAA destruction by the ubiquitin-proteasome system to activate auxin-response gene expression (*Tan et al., 2007*). Unlike conventional small molecules that bind to active sites or allosteric sites to modulate the activity of their target proteins, molecular glues influence the activity or fate of their target proteins by bringing them to the vicinity of regulatory proteins. As a result, molecular glues have been viewed enthusiastically as a unique pharmacological modality to target proteins without druggable pockets (*Maniaci and Ciulli, 2019*).

Notwithstanding, molecular glues are rare and were discovered serendipitously; only a handful of molecular glues have been documented over the past four decades. For example, macrocyclic natural products cyclosporin A, FK506, and rapamycin recruit FKBP proteins to the phosphatase calcineurin or the kinase mTORC1, interfering with their enzymatic activities to control intracellular signal transduction (*Brown et al., 1994*; *Liu et al., 1991*; *Sabatini et al., 1994*). Similarly, the thalidomide class of immunomodulatory drugs binds to the E3 ubiquitin ligase cereblon and alters its substrate

specificity to recognize and degrade several zinc finger transcription factors (*Ito et al., 2010*; *Krönke et al., 2014*; *Lu et al., 2014*). More recently, indisulam and related anticancer sulfonamides were discovered to function by promoting the interaction between the splicing factor RBM39 and the E3 ubiquitin ligase DCAF15, resulting in the degradation of RBM39 to cause aberrant pre-mRNA splicing (*Han et al., 2017*; *Uehara et al., 2017*).

Several common themes have emerged from this limited set of molecular glues. First, molecular glues often bind to their target proteins with modest or even undetectable affinities, while enhanced affinities are usually observed once a regulatory protein is also present in the system (*Bussiere et al., 2020*; *Du et al., 2019*; *Faust et al., 2020*). Such unique binding characteristics can be explained by induced protein-protein interactions and pre-existing protein surface complementarity as revealed by structural analysis of molecular glue-bound protein complexes (*Bussiere et al., 2020*; *Du et al., 2019*; *Faust et al., 2020*; *Tan et al., 2007*). Second, molecular glues can target transcription factors and splicing factors, by recognizing either relatively flat surfaces or disordered regions of these proteins (*Han et al., 2017*; *Krönke et al., 2014*; *Lu et al., 2014*; *Tan et al., 2007*). Third, molecular glues possess favorable pharmacological properties to serve as drug candidates. For example, the compactness of the thalidomide scaffold allowed the development of bivalent proteolysis targeting chimeras to direct cereblon to degrade disease-relevant proteins (*Lu et al., 2015*; *Winter et al., 2015*). However, the rarity of molecular glues has limited their potential as a general strategy for drug development.

In this study, we report the discovery and the mechanism of action of a new molecular glue HQ461. HQ461 was identified through phenotype-based high-throughput small-molecule screening and found to possess potent cytotoxicity. Combining chemical genetics and biochemical reconstitution, we show that HQ461 acts by binding to CDK12's kinase domain, creating a modified CDK12 surface to recruit DDB1, which is a subunit of the DDB1-CUL4-RBX1 E3 ubiquitin ligase. Distinct from existing molecular glues that engage the ubiquitin proteasome system by binding to a substrate-specific receptor, HQ461 converts CDK12 into a substrate-specific receptor for DDB1-CUL4-RBX1 to trigger the polyubiquitination and subsequent degradation of CDK12's partner protein Cyclin K (CCNK). HQ461-mediated depletion of CCNK from cells leads to reduced Serine 2 phosphorylation of RNA polymerase II C-terminal domain (CTD) and affects the expression of genes involved in DNA damage response.

## Results

### HQ461's cytotoxicity requires DDB1-CUL4-RBX1

HQ461 (*Figure 1A*) is a compound originated from phenotype-based high-throughput screening for small molecules that suppress NRF2 activity (*Figure 1—figure supplement 1*). Phenotypic characterization of HQ461 revealed its potent cytotoxicity to the non-small cell lung cancer cell line A549 with a half-maximal inhibitory (IC$_{50}$) concentration of 1.3 µM (*Figure 1B* and *Figure 1—source data 1*). To investigate HQ461's mechanism of action, we performed pooled genome-wide CRISPR-Cas9 knockout screening by targeting 19,114 genes with four individual sgRNAs per gene (*Doench et al., 2016*; *Figure 1—figure supplement 2A*). We treated sgRNA-transduced A549 cells with either the vehicle DMSO or a sub-lethal dose of HQ461 for 3 weeks to enrich for cells that were resistant to HQ461's cytotoxic effects. Afterwards, we isolated genomic DNAs from surviving cells and performed sgRNA amplification by PCR followed by next-generation sequencing to measure the abundance of each sgRNA in these cells. Using the MAGeCK algorithm (*Li et al., 2014*) to rank enriched genes in HQ461 treated relative to DMSO-treated cells, we found the top-ranking genes include the E2 ubiquitin activating enzyme G1 (*UBE2G1*), five members of the DDB1-CUL4-RBX1 E3 ubiquitin ligase complex (*DDB1*, *RBX1*, *DDA1*, *CUL4A*, *CUL4B*), and multiple regulators of DDB1-CUL4-RBX1 activity (*GLMN*, *NAA30/35/38*, *NAE1*, and *UBE2M*) (*Figure 1C*, *Figure 1—figure supplement 2B*, and *Figure 1—source data 2*). We targeted three top-ranking genes *DDB1*, *RBX1*, and *UBE2G1* individually by two independent sgRNAs; expression of these sgRNAs in A549 cells resulted in the depletion of their target proteins (*Figure 1—figure supplement 2C*) and resistance to HQ461's toxicity (*Figure 1D*, *Figure 1—source data 3* and *Figure 1—figure supplement 2D*). Because all these candidate genes encode proteins in the ubiquitin proteasome system, we hypothesized that HQ461 may exert its cytotoxicity by triggering proteasomal degradation of target protein(s).

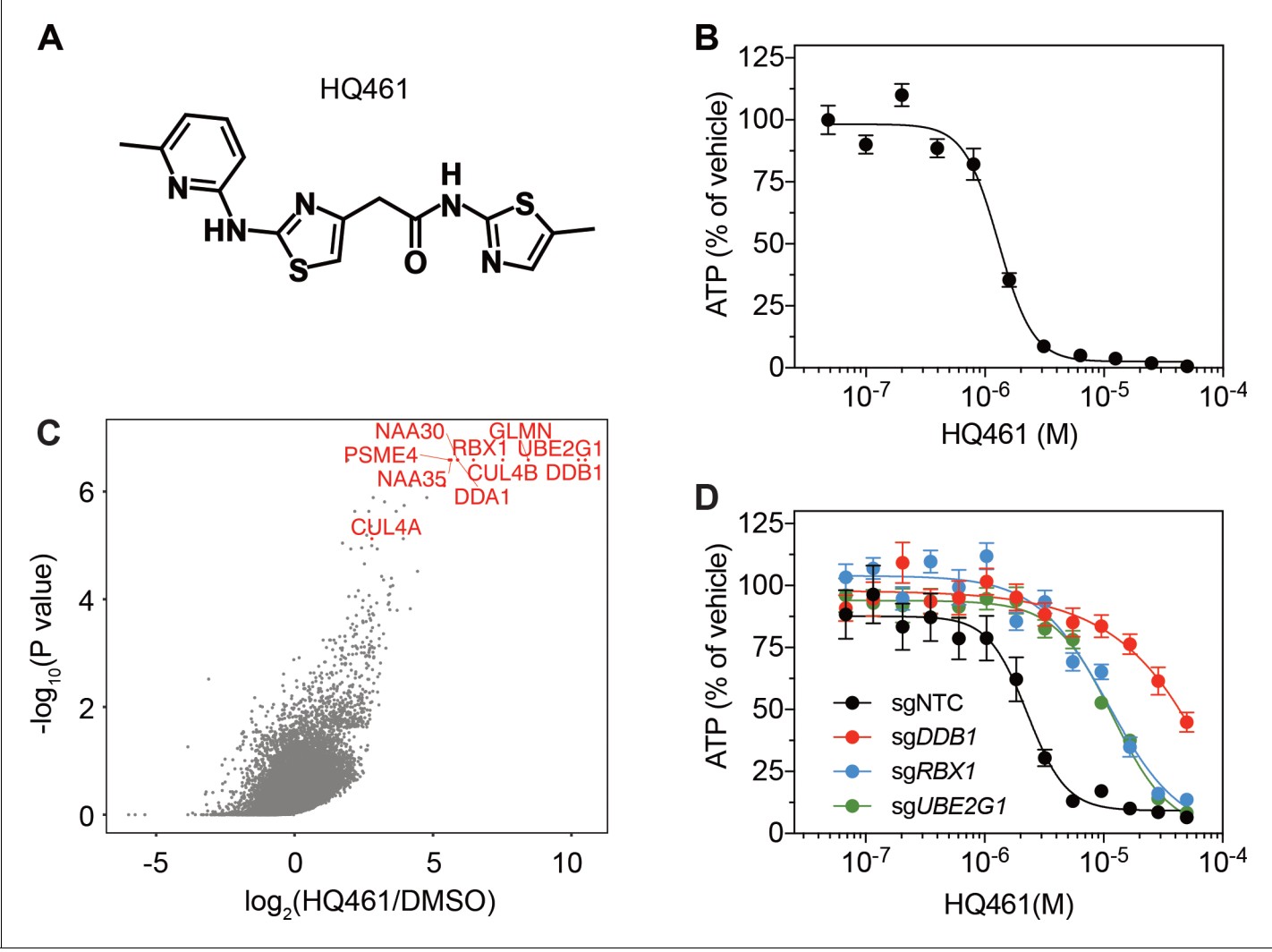

**Figure 1.** DDB1-CUL4-RBX1 mediates HQ461's cytotoxicity. (**A**) Chemical structure of HQ461. (**B**) Measurement of the HQ461 IC$_{50}$ on the viability of A549 cells (IC$_{50}$ = 1.3 μM, 95% confidence interval (CI): 1.0 μM-1.6 μM). Error bars represent standard errors of mean (SEM) from three biological replicates. (**C**) MAGeCK analysis of pooled genome-wide CRISPR-Cas9 sgRNA screening of HQ461 resistance in A549 cells. (**D**) Measurement of the HQ461 IC$_{50}$ on the viability of A549 cells expressing non-targeting control (NTC, IC$_{50}$ = 2.3 μM, 95% CI: 1.9 μM-2.8 μM) or sgRNAs targeting *DDB1* (IC$_{50}$ >28.8 μM), *RBX1* (IC$_{50}$ = 10.9 μM, 95% CI: 6.5 μM-101 μM), or *UBE2G1* (IC$_{50}$ = 11.9 μM, 95% CI: 9.8 μM-15.5 μM). Error bars represent SEM from three biological replicates.

The online version of this article includes the following source data and figure supplement(s) for figure 1:

**Source data 1.** Measurement of the HQ461 IC$_{50}$ on the viability of A549 cells (Source data for *Figure 1B*).
**Source data 2.** MAGeCK analysis of pooled genome-wide CRISPR-Cas9 sgRNA screening of HQ461 resistance in A549 cells (Source data for *Figure 1C*).
**Source data 3.** Measurement of the HQ461 IC$_{50}$ on the viability of A549 cells expressing non-targeting control or sgRNAs targeting DDB1, RBX1, or UBE2G1 (Source data for *Figure 1D*).
**Figure supplement 1.** High-throughput chemical screening of NRF2 inhibitors.
**Figure supplement 2.** CRISPR-Cas9 screening for HQ461 resistance.

## *CDK12* mutations cause HQ461 resistance

To identify any target protein(s) destabilized by HQ461, we performed gain-of-function genetic screening in the colorectal cancer cell line HCT-116. HCT-116 cells are defective in mismatch repair, and therefore exhibits a high rate of random point mutations (*Han et al., 2016*; *Wacker et al., 2012*). Our previous studies suggest that the identification of recurrent mutations present in multiple independent drug-resistant clones may reveal the direct drug target (*Han et al., 2016*; *Han et al., 2017*). To ensure the isolation of independent HQ461-resistant clones, we derived 10 clonal isolates

from HCT-116 that were initially sensitive to HQ461 (HQ461$^S$). Five of these clones evolved HQ461 resistance (HQ461$^R$) during the expansion from a single cell into a large cell population (*Figure 2A*). We isolated one HQ461$^R$ clone from each cell population and confirmed their resistance to HQ461 (three-fold increase of IC$_{50}$ relative to parental HCT-116 cells) (*Figure 2B* and *Figure 2—source data 1*). We pooled genomic DNAs isolated from the five HQ461$^R$ clones at equal amounts for whole-exome sequencing. As a control, we sequenced the exomes of their corresponding HQ461$^S$ clones as a pool.

We then used the GATK pipeline (*DePristo et al., 2011*; *McKenna et al., 2010*; *Van der Auwera et al., 2013*) to identify variants that were unique to the HQ461$^R$ group. The top-ranking variant was a G to A substitution resulting in a non-synonymous substitution of glycine to glutamate (G731E) in the gene Cyclin-dependent kinase 12 (*CDK12*). This variant occurred at an allele frequency of ~30% in HQ461$^R$ versus 0% in HQ461$^S$ (*Figure 2C* and *Figure 2—source data 2*). Deconvolution of pooled samples by Sanger sequencing of *CDK12* revealed a heterozygous *CDK12* G731E mutation in three out of five clones, consistent with an overall allele frequency of 30% in a diploid cell line (*Figure 2C*). Using Sanger sequencing of *CDK12* to examine additional HQ461$^R$ clones, we discovered another *CDK12* mutation affecting the same codon which replaces glycine with arginine (G731R) (*Figure 2C*).

To test if G731E and G731R mutations in *CDK12* confer HQ461 resistance, we knocked-in these two mutations in A549 cells using CRISPR-Cas9 genome editing. We designed an sgRNA targeting a 21-nucleotide sequence upstream of codon G731 of *CDK12*. G731 is encoded by the codon GGG, serving conveniently as a protospacer adjacent motif (PAM) for Cas9 (*Jinek et al., 2012*). To distinguish knock-in mutations from spontaneous mutations, we designed two-nucleotide substitutions on the repair templates to encode G731E (GGG to GAA) or G731R (GGG to CGC) (*Figure 2D*). Successful editing of GGG sequence to GAA or CGC destroys the PAM sequence, preventing the Cas9/sgRNA complex from re-cutting.

We co-transfected plasmids expressing Cas9, the *CDK12*-targeting sgRNA, and single-stranded oligodeoxynucleotides (ssODNs) encoding the G731E or G731R allele. These conditions increased clonal resistance to HQ461 (*Figure 2E*). Four independent clones were isolated. They contained the expected G731E or G731R mutation (*Figure 2—figure supplement 1A*) and were between 9- and 13-fold less sensitive to HQ461 than parental A549 cells (*Figure 2—figure supplement 1B*). Extending from these observations, we generated stable cell lines expressing either wild-type CDK12 or CDK12 carrying G731E or G731R substitution. Whereas expression of wild-type CDK12 did not alter A549's sensitivity to HQ461, expression of CDK12 G731E or G731R resulted in HQ461 resistance (*Figure 2—figure supplement 1C*).

## HQ461 triggers polyubiquitination and proteasomal degradation of CCNK

CDK12 is a large protein with a central kinase domain flanked by an arginine serine rich (RS) domain and two proline rich motifs. The G731 hotspot for HQ461 resistance mutations is located in the kinase domain of CDK12 (*Figure 2F*). This structural position for the mutation led us to speculate that CDK12 might be a protein substrate destabilized by DDB1-CUL4-RBX1 following HQ461 treatment and that G731E or G731R substitution in CDK12 might interfere with such destabilization. By monitoring the protein level of CDK12 at different time points of HQ461 treatment, we found a modest 50% reduction of CDK12 protein level after 8 hr of treatment (*Figure 3A*).

To examine if a 50% reduction of CDK12 protein might help explain HQ461's cytotoxicity, we used CRISPR interference (CRISPRi) (*Gilbert et al., 2014*) to downregulate *CDK12* transcription. We used 10 independent sgRNAs targeting the transcription start site of *CDK12* and found all of them efficiently depleted CDK12 mRNA and protein (between 5- and 10-fold reduction) without causing cell death or a proliferation defect (*Figure 3—figure supplement 1A*). This result suggested that modest CDK12 downregulation by HQ461 is not a direct cause of cell death. CDK12 binds to Cyclin K (CCNK) to form a functional and active complex to perform its functions (*Blazek et al., 2011*; *Cheng et al., 2012*). In contrast to the modest effect on CDK12 protein, HQ461 treatment led to >8 fold reduction of the CCNK protein level after a short 4 hr treatment (*Figure 3A*). In cells with CDK12 G731E or G731R mutation, CCNK protein was not affected by HQ461 treatment (*Figure 3B*). The effect of HQ461 on CDK12 protein level is likely a result of CCNK depletion because CRISPR inactivation of CCNK also resulted in a reduction of CDK12 protein (*Figure 3—figure supplement 1B*).

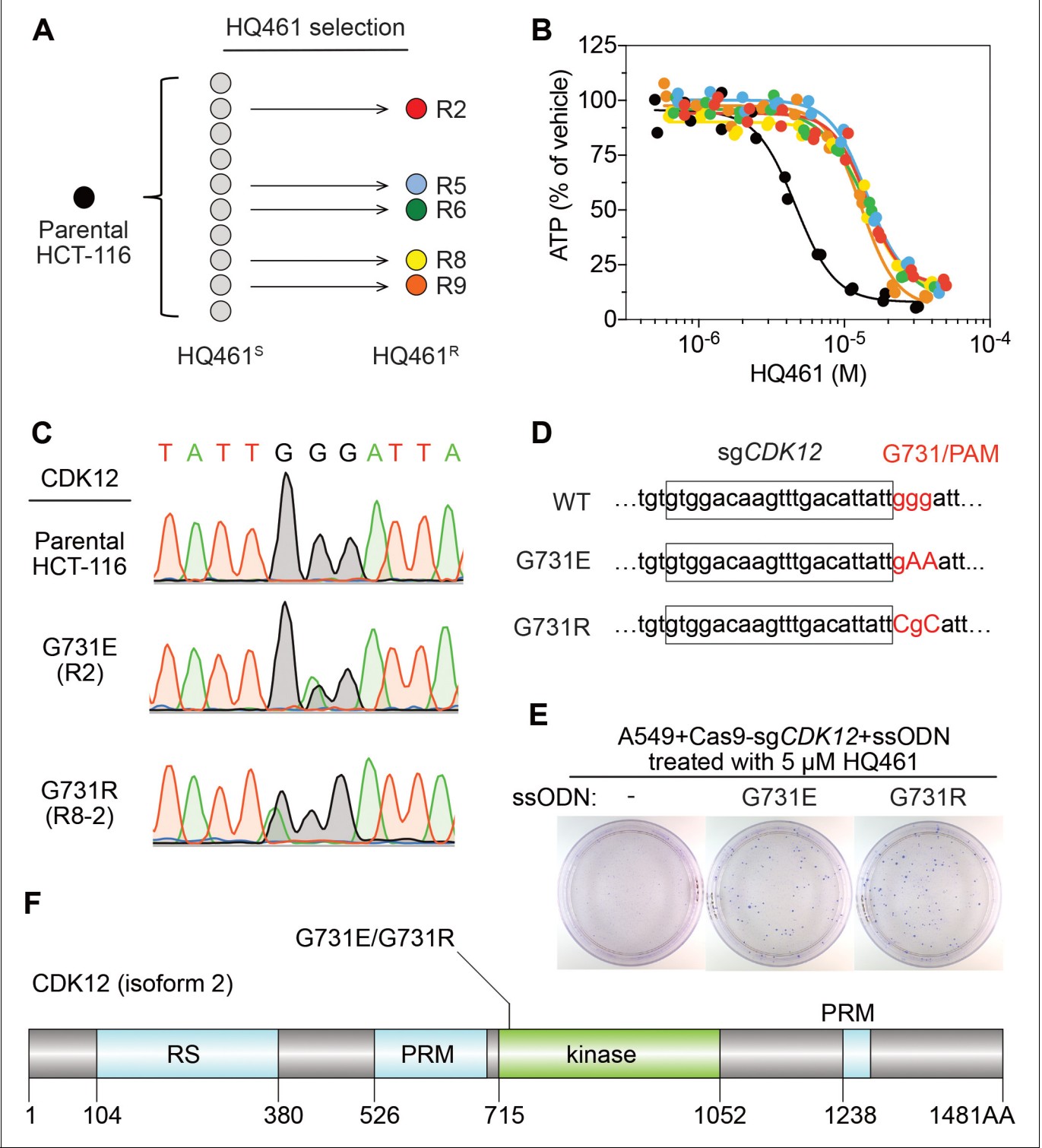

**Figure 2.** Mutations in CDK12 confer resistance to HQ461. (**A**) Strategy for isolation of independent HQ461-resistant HCT-116 clones. (**B**) Measurement of the HQ461 $IC_{50}$ on the viability of parental HCT-116 ($IC_{50}$ = 4.6 µM, 95% CI: 4.1 µM-5.3 µM) and five HQ461-resistant HCT-116 clones (color coded in (**A**), $IC_{50}$ ranging from 13.0 µM to 14.3 µM). Each point represents the average of two biological replicates. (**C**) Sanger sequencing verification of *CDK12* mutations found in HQ461-resistant HCT-116 clones. (**D**) The edited genomic sequence of *CDK12*, highlighting the PAM motif, the target sequence (boxed), and G731E or G731R mutations. (**E**) *CDK12* G731E or G731R knock-in in A549 cells gives rise to HQ461 resistance, visualized by crystal violet staining. Mock transfection (omitting the repair template) results in no HQ461 resistance. (**F**) The domain structure of CDK12 (isoform two is shown here and used in this study). G731 is located within the central kinase domain of CDK12.

*Figure 2 continued on next page*

*Figure 2 continued*

The online version of this article includes the following source data and figure supplement(s) for figure 2:

**Source data 1.** Measurement of the HQ461 IC$_{50}$ on the viability of parental HCT-116 and five HQ461-resistant HCT-116 clones (source data for *Figure 2B*).
**Source data 2.** Exome-sequencing of HQ461S versus HQ461R HCT-116 (source data for *Figure 2C*).
**Figure supplement 1.** The *CDK12* G731E and G731R mutations confer HQ461 resistance.

To test if CCNK depletion requires a functional ubiquitin proteasomal system, we used a proteasome inhibitor bortezomib (*Goldberg, 2012*) and a neddylation inhibitor MLN4924 (*Soucy et al., 2009*) (inactivating all Cullin-RING ligases including DDB1-CUL4-RBX1) and found both inhibitors abrogated HQ461-dependent CCNK degradation (*Figure 3C*). Furthermore, in cells depleted of DDB1 or RBX1, CCNK remained stable following HQ461 treatment (*Figure 3—figure supplement 1C*). We next sought to examine if HQ461-triggered CCNK polyubiquitination, so we co-transfected HEK293T cells with plasmids encoding 6xHis-tagged ubiquitin, 3xHA-tagged CCNK, and 3xFLAG-tagged CDK12 wild-type or mutants with G731E/R substitution. After purification of lysates by nickel chromatography, we performed western blotting with anti-HA and anti-FLAG antibodies to examine if CDK12 and CCNK were polyubiquitinated in cells treated with HQ461. We observed increased CCNK polyubiquitination following HQ461 treatment when wild-type CDK12 was co-expressed. In contrast, co-expression of CDK12 G731E or G731R impeded CCNK polyubiquitination. Furthermore, neither wild-type or mutant CDK12 was polyubiquitinated (*Figure 3D*). Taken together, these results support the conclusion that HQ461 triggers CCNK polyubiquitination as mediated by DDB1-CUL4-RBX1 and that G731E/R mutations in CDK12 prevent CCNK polyubiquitination.

CDK12 is known to interact with CCNK via its kinase domain (*Bösken et al., 2014*; *Dixon-Clarke et al., 2015*). Co-expression of CDK12's kinase domain with CCNK was sufficient to trigger both CCNK polyubiquitination and degradation (*Figure 3E* and *Figure 3—figure supplement 1D*). In contrast, expression of CDK12's kinase domain with G731E/R mutations blocked CCNK polyubiquitination and degradation (*Figure 3E* and *Figure 3—figure supplement 1D*). CDK13 is a paralog of CDK12 with a similar domain architecture and its kinase domain shares 90.5% identity to CDK12's kinase domain (*Greifenberg et al., 2016*). Expression of the wild-type kinase domain of CDK13 also supported CCNK degradation. Mutating CDK13's glycine 709 to glutamate or arginine (G709E or G709R), which is located at an equivalent position of glycine 731 in CDK12, also blocked CCNK degradation (*Figure 3—figure supplement 1E*).

## HQ461 mediates interaction between CDK12 and DDB1

To characterize the mechanisms through which HQ461 triggers CCNK degradation, we used CRISPR/Cas9 to knock-in a sequence encoding an N-terminal 3xFLAG tag into the endogenous *CDK12* locus and then immunoprecipitated 3xFLAG-CDK12 and its interacting proteins from cellular lysates treated with HQ461. Using western blotting, we found CDK12 stably associated with CCNK, doing so in an HQ461-independent manner. In contrast, DDB1 was only detected with the CDK12/CCNK complex isolated from lysates treated with HQ461 (*Figure 4A*), suggesting that HQ461 mediates recruitment of CDK12/CCNK to DDB1. Furthermore, using A549 cell lines stably expressing 3xFLAG-CDK12 wild-type, G731E, or G731R, we found that G731E or G731R impeded HQ461-mediated recruitment of CDK12/CCNK to DDB1 (*Figure 4B*).

To recognize diverse substrates, DDB1-CUL4-RBX1 requires a type of substrate-specific receptor proteins known as DDB1 CUL4 associated factors (DCAF). More than 60 human proteins have been proposed as DCAFs (*Lee and Zhou, 2007*) and fewer than 30 are likely bona fide DCAFs. However, our genome-wide CRISPR-Cas9 screening of HQ461 resistance did not reveal any candidate DCAF (*Figure 1C* and *Figure 1—figure supplement 2B*), suggesting that HQ461-dependent degradation of CCNK may not require a DCAF. To test this hypothesis, we next reconstituted CCNK polyubiquitination in vitro with purified recombinant ubiquitin, E1 (UBA1), E2 (UBE2G1 and UBE2D3), and E3 (CUL4-RBX1-DDB1) (*Figure 4C*). We observed polyubiquitination of CCNK but not CDK12 in the presence of E1, E2, E3, and HQ461 in vitro; omitting any enzyme or HQ461 from the reaction prevented CCNK polyubiquitination (*Figure 4—figure supplement 1*). CCNK in complex with CDK12 G731E or G731R was resistant to HQ461-induced polyubiquitination (*Figure 4D*). Taken together,

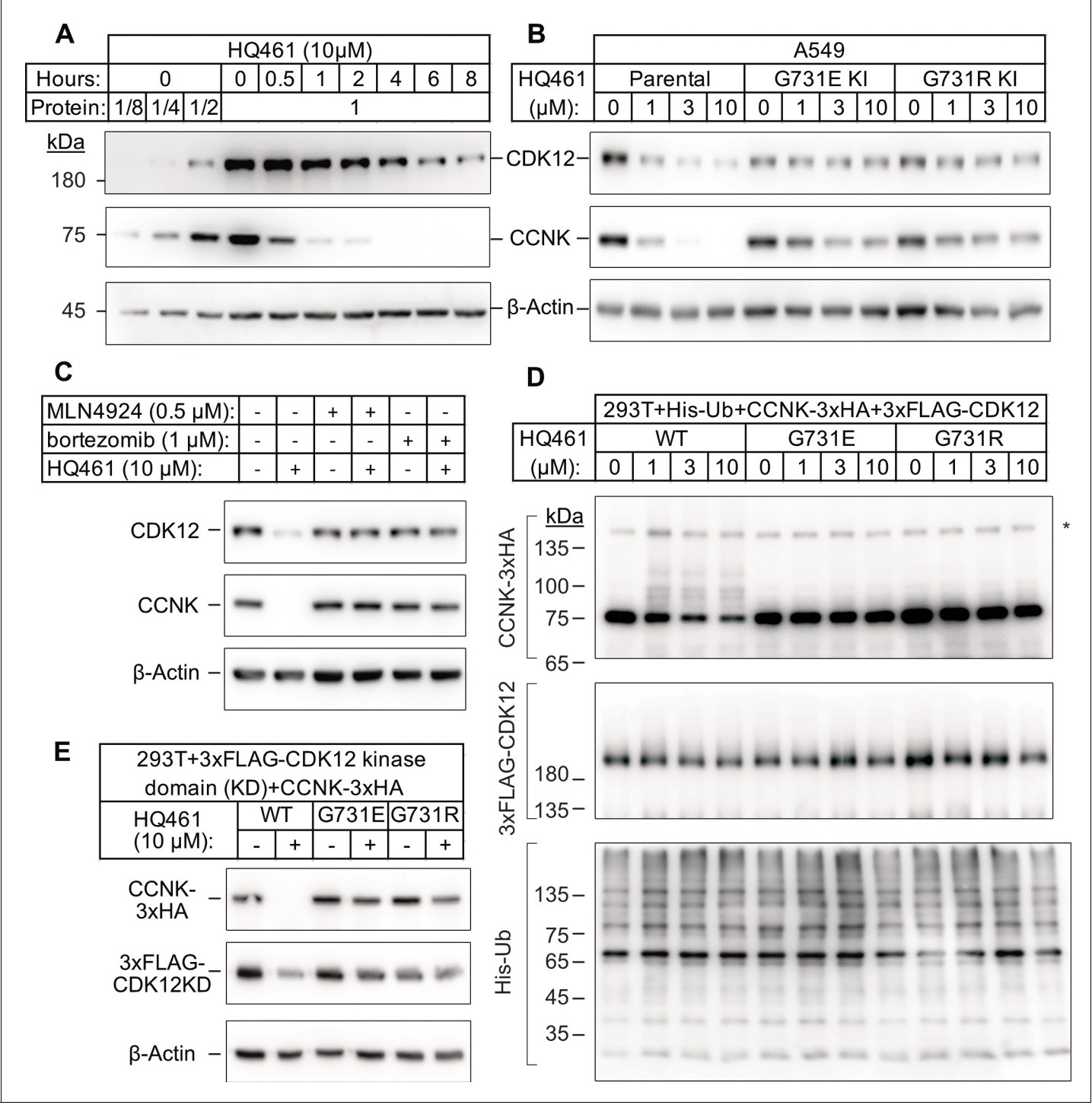

**Figure 3.** HQ461 promotes CCNK polyubiquitination and degradation through DDB1-CUL4-RBX1. (**A**) Western blotting of CDK12 and CCNK from A549 cells treated with HQ461. Dilutions of untreated sample (0 hr) were included to allow quantitative assessment of the reduction of protein levels. (**B**) Effect of *CDK12* G731E and G731R mutations on CCNK and CDK12 protein levels in A549 cells treated with HQ461. (**C**) Effect of bortezomib and MLN4924 on CCNK and CDK12 protein levels in A549 cells treated with HQ461. (**D**) HQ461 triggers in vivo polyubiquitination of CCNK in complex with wild-type CDK12 but not CDK12 G731E or G731R. The asterisk indicates a non-specific band. (**E**) Wild-type CDK12 kinase domain but not G731E or G731R mutant is sufficient for mediating HQ461-dependent destabilization of CCNK.

The online version of this article includes the following figure supplement(s) for figure 3:

**Figure supplement 1.** Both CDK12 and CDK13 support HQ461-induced CCNK degradation.

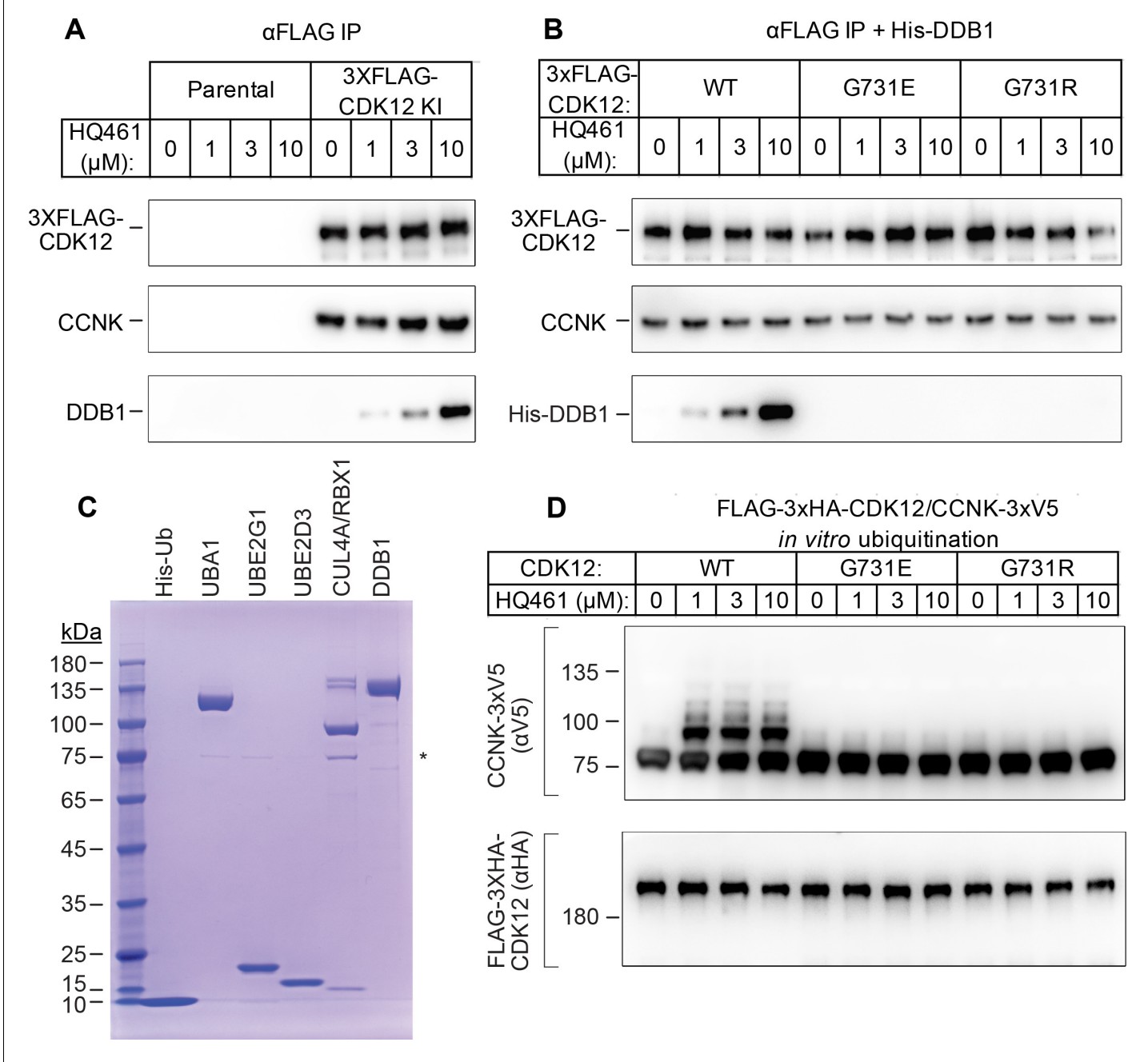

**Figure 4.** HQ461 functions as a molecular glue between CDK12 and DDB1 to promote CCNK polyubiquitination. (**A**) Coimmunoprecipitation of DDB1 with 3xFLAG-CDK12/CCNK from A549 lysates treated with HQ461. (**B**) Coimmunoprecipitation of His-DDB1 with 3xFLAG-CDK12/CCNK WT, G731E, or G731R from A549 lysates treated with HQ461. (**C**) Coomassie blue staining of recombinant proteins used for an in vitro ubiquitination assay. The asterisk indicates contaminating HSP70. (**D**) In vitro ubiquitination of CDK12/CCNK with recombinant ubiquitin, E1 (UBA1), E2 (UBE2G1 and UBE2D3), and E3 (DDB1-CUL4-RBX1).

The online version of this article includes the following figure supplement(s) for figure 4:

**Figure supplement 1.** In vitro polyubiquitination of CDK12/CCNK.

these results suggest that HQ461 promotes the recruitment of CDK12/CCNK to DDB1-CUL4-RBX1 for polyubiquitination.

## HQ461 functions as a molecular glue between CDK12 and DDB1

The apparent biochemical activity of HQ461 in mediating protein-protein interactions resembles previously reported molecular glues such as auxin, lenalidomide, and indisulam (*Han et al., 2017*; *Krönke et al., 2014*; *Lu et al., 2014*; *Tan et al., 2007*; *Uehara et al., 2017*). To test if HQ461 also functions as a molecular glue, we purified FLAG-DDB1 as well as CDK12's kinase domain (CDK12KD) in complex with the Cyclin box domain of Cyclin K (CCNKΔC). Using FLAG antibody-conjugated resin to pulldown FLAG-DDB1, we found that DDB1 formed a near-stoichiometric complex with CDK12KD/CCNKΔC in an HQ461-dependent manner (*Figure 5—figure supplement 1A*). We further developed an AlphaScreen assay to measure the interaction between CDK12KD/CCNKΔC and DDB1, and observed that HQ461 induced the formation of a CDK12KD/CCNKΔC/DDB1 complex with an apparent $EC_{50}$ of 1.9 μM. G731E and G731R mutants of CDK12KD/CCNKΔC did not form a complex with DDB1 even with 20 μM of HQ461 (*Figure 5A* and *Figure 5—source data 1*).

To map the HQ461-induced protein-protein interaction interface, we performed chemical cross-linking mass spectrometry (CXMS) (*Liu and Heck, 2015*). Specifically, the HQ461-induced CDK12KD/CCNKΔC/DDB1 complex was cross-linked using two lysine-specific cross-linkers (DSS and BS3), followed by digestion with trypsin and LC-MS/MS analysis. Using pLink 2 software (*Chen et al., 2019*), 41 pairs of cross-linked peptides were identified, out of which only one pair was derived from inter-protein cross-linking; the remainder were all derived from intra-protein cross-linking. For this single inter-protein peptide pair, lysine 745 of CDK12 was cross-linked to lysine 867 of DDB1 (*Figure 5B*, *Figure 5—figure supplement 1B*, and *Figure 5—source data 2*). This CXMS result suggests that HQ461 induces the formation of an interface between CDK12's kinase domain and DDB1.

Like other kinases, CDK12's kinase domain possesses an ATP-binding pocket sandwiched between an N-terminal β sheet-rich lobe and a C-terminal α helix-rich lobe (*Bösken et al., 2014*; *Dixon-Clarke et al., 2015*). Both lysine 745 and glycine 731 (HQ461-resistant mutation hotspot) reside in the N-terminal lobe atop the ATP-binding pocket of CDK12, suggesting that HQ461 may bind to this pocket, creating a modified CDK12 surface to bind to DDB1 (*Figure 5B*). We tested this hypothesis by directly measuring HQ461 binding to CDK12KD/CCNKΔC using differential scanning fluorimetry (DSF) with a fluorescent dye Sypro Orange. We monitored the thermal denaturation of CDK12KD/CCNKΔC in the presence or absence of HQ461 and found that HQ461 stabilized CDK12KD/CCNKΔC by a 1.4°C increase in its melting temperature (Tm), consistent with the model that HQ461 directly binds to CDK12KD/CCNKΔC (*Figure 5C*).

DDB1 is composed of three β propeller domains (BPA, BPB, and BPC) (*Figure 5B*; *Li et al., 2006*). The residue lysine 867 on DDB1 that was cross-linked to CDK12 in the CXMS experiment is located in the BPC domain of DDB1, which is known to interact with a promiscuous α-helical motif present in DCAF proteins and viral proteins such as the hepatitis B virus X protein (HBx) (*Figure 5B*; *Li et al., 2010*). We found that a peptide corresponding to the α-helical motif of HBx increased the thermal stability of recombinant DDB1 without its BPB domain (DDB1ΔBPB) by a 2°C increase in Tm (*Figure 5D*). In contrast, HQ461 did not change the thermal stability of DDB1ΔBPB (*Figure 5D*). These results suggest that HQ461 does not directly bind to DDB1.

We next evaluated whether HQ461's binding to CDK12KD/CCNKΔC was affected by G731E/R mutations. Using label-free nanoDSF, we monitored thermal unfolding of CDK12KD/CCNKΔC (wild type or G731E/R mutants) in the presence of increasing concentrations of HQ461. Tm of all three protein preparations increased in an HQ461 concentration-dependent manner (*Figure 5E*), suggests that HQ461 is capable of binding to CDK12's kinase domain with G731/R mutations.

THZ531 (*Figure 5—figure supplement 2A*) is a specific covalent inhibitor of CDK12/CDK13 by irreversibly targeting a cysteine uniquely present in CDK12/13's kinase domains (*Figure 4D*; *Zhang et al., 2016*). Using the AlphaScreen assay, we found that THZ531 inhibited HQ461-dependent formation of the CDK12KD/CCNKΔC/DDB1 complex (*Figure 5A*). Dinaciclib (Figure S6B) is a non-covalent CDK inhibitor targeting CDK1, 2, 5, 9, 12, 13 (*Johnson et al., 2016*). Neither THZ531 nor dinaciclib treatment caused CCNK degradation (*Figure 5—figure supplement 2B*). In contrast, treating cells with THZ531 or dinaciclib prior to HQ461 treatment prevented CCNK degradation (*Figure 5—figure supplement 2C*), clearly suggesting that THZ531, dinaciclib, and HQ461 occupy the same ATP-binding pocket in CDK12's kinase domain. These results confirm HQ461's engagement with CDK12's kinase domain in vivo, and highlight HQ461's unique ability to induce CCNK degradation.

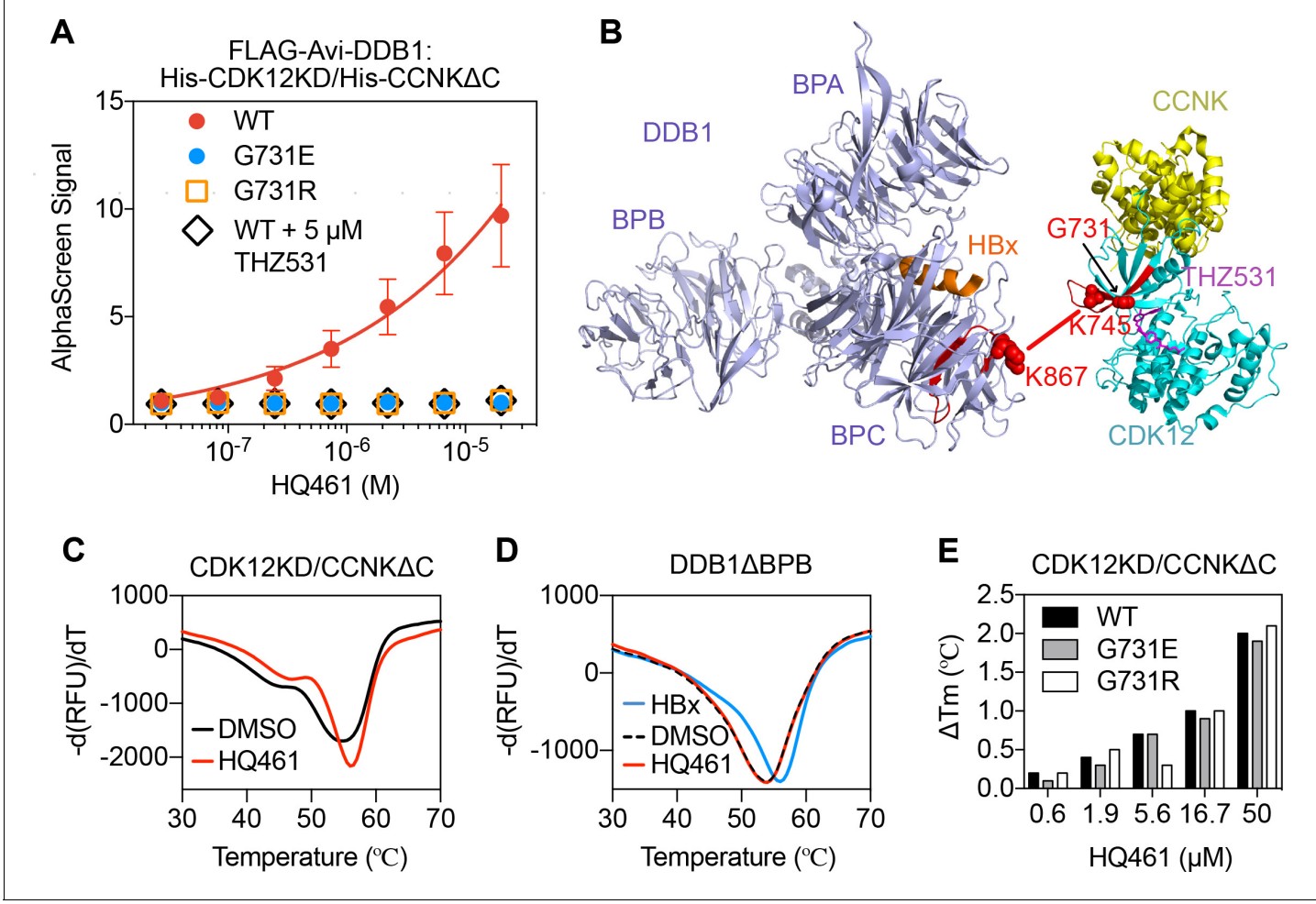

**Figure 5.** HQ461 binds to CDK12 to recruit DDB1. (**A**) Detection of HQ461-dependent interaction between FLAG-Avi-DDB1 and His-CDK12KD/His-CCNKΔC examined in an AlphaScreen assay. Error bars represent SEM from three technical replicates. (**B**) CXMS analysis identified one inter-protein cross-link between the DDB1 (PDB code: 3i7h) lysine 867 and CDK12KD/CCNKΔC (PDB code: 5acb) lysine 745. (**C**) DSF analysis of CDK12KD/CCNKΔC thermal unfolding in the presence of 50 µM HQ461 or vehicle DMSO. (**D**) DSF analysis of DDB1ΔBPB thermal unfolding in the presence of DMSO, 50 µM HQ461 or HBx peptide. (**E**) Nano DSF analysis of CDK12KD/CCNKΔC (WT or G731E/R) thermal unfolding in the presence of increasing concentrations of HQ461. Δ Tm is calculated by subtracting the Tm of DMSO control.

The online version of this article includes the following source data and figure supplement(s) for figure 5:

**Source data 1.** Detection of HQ461-dependent interaction between FLAG-Avi-DDB1 and His-CDK12KD/His-CCNKΔC examined in an AlphaScreen assay (source data for *Figure 5A*).

**Source data 2.** CXMS analysis of DDB1 and CDK12KD/CCNKΔC in the presence of HQ461 (source data for *Figure 5B*).

**Figure supplement 1.** Biochemical characterization of HQ461's molecular-glue activity.

**Figure supplement 2.** HQ461 binds to the ATP-binding pocket of CDK12.

## Structure-activity relationship study of HQ461

To dissect the chemical basis for optimizing HQ461's molecular-glue activity, we conducted a thorough structure-activity relationship study of HQ461 (*Figure 6* and *Figure 6—figure supplement 1*). We made a CCNK-*luc* reporter that converts the degradation of CCNK into a reduction in luciferase activity (*Figure 6—figure supplement 2*). We then measured the ability of synthesized HQ461 analogs in reducing CCNK-*luc* activity at 10 µM relative to DMSO control (maximal degradation, Dmax) (*Figure 6—figure supplement 1*). For analogs with a Dmax greater than 50%, we further performed dose response analysis to measure their half-maximal CCNK-degradation concentrations (DC$_{50}$) (*Figure 6* and *Figure 6—figure supplement 2*).

| ID | R1 | DC50 (µM) | ID | R1 | DC50 (µM) |
|---|---|---|---|---|---|
| HQ461 | N—Me | 0.132 | HQ006 | N—OMe | 0.466 |
| HQ001 | N—F | 0.300 | HQ007 | N—NH2 | 0.561 |
| HQ002 | N—Cl | 0.471 | HQ008 | N—N(pyrazole) | 5.030 |
| HQ003 | N—Br | 0.407 | HQ009 | N—CF3 | 2.357 |
| HQ004 | N—Et | 0.542 | HQ010 | N (pyridine) | 0.811 |
| HQ005 | N—CH2OH | 0.041 | HQ011 | H | NA |

**Figure 6.** Structure-activity relationship of HQ461 analogs. Half maximal CCNK-degradation concentrations are reported for each analog. CCNK degradation was measured using a CCNK-*luc* reporter.

The online version of this article includes the following figure supplement(s) for figure 6:

**Figure supplement 1.** Structure-activity relationship of HQ461 analogs.

**Figure supplement 2.** Dose response curves of HQ461 analogs in reducing CCNK-*luc* activity.

We first modified the aminopyridinylthiazol scaffold either by changing pyridine to other substituted heterocycles or by elongating the linker to increase the overall size of the molecule (*Figure 6— figure supplement 1*). Specifically, we changed the pyridine to small-size groups such as hydrogen (HQ011) and acetyl (HQ019), or replaced the linker between pyridine and aminothiazol to amide

(HQ014). All these modifications resulted in losses in activity. We also made changes to the 5-methylthiazol-2-amine moiety of HQ461 by incorporating several substitutions (HQ015, 016, and 017) and discovered that this moiety was crucial for activity. We therefore maintained this pharmacophore and incorporated a variety of substitutions on the pyridine ring (*Figure 6*). The elimination of the methyl group (HQ010) on the pyridine ring resulted in a 6-fold higher $DC_{50}$ value, suggesting an important steric effect of the 5-position on the pyridine ring. Consistent with this hypothesis, the incorporation of a bulky ring structures on pyridine's 5-position (HQ008) led to a 38-fold increase in $DC_{50}$. In addition, substitution of the 5-position with F, Cl, Br, ethyl, methoxyl, and trifluoromethyl (HQ001-004, 006, and 009) all diminished potency.

Finally, a beneficial effect of polarity is observed by a 3-fold increase in potency of HQ005 ($DC_{50}$: 41 nM) in which the methyl group in HQ461 was replaced by a hydroxymethyl group. We speculate that the hydroxyl group may interact favorably with polar residues in CDK12's kinase domain.

## HQ461-dependent degradation of CCNK causes cell death via inactivation of CDK12/13

We next examined if HQ461-induced CCNK depletion is the cause of cell death. Using a colony formation assay, expression of an sgRNA targeting *CCNK* in A549 cells resulted in reduced cell viability. The inhibitory effect of this *CCNK*-targeting sgRNA could be reversed by expressing a cDNA sequence encoding an sgRNA-resistant variant of *CCNK*, confirming that the observed effect results from CCNK inactivation (*Figure 7A*).

CCNK forms two distinct complexes with CDK12 and with CDK13, and is essential for the kinase activity of both complexes (*Blazek et al., 2011*; *Cheng et al., 2012*). Inactivation of CCNK therefore blocks the functions of both CDK12 and CDK13. The C-terminal domain (CTD) of RNA polymerase II's largest subunit is composed of 52 repeats with the consensus heptad sequence YSPTSPS. Serine 2 of the heptad repeat is a known phosphosubstrate of CDK12/13 (*Blazek et al., 2011*). HQ461 treatment reduced POLII CTD Serine 2 phosphorylation without affecting the level of total POLII CTD (*Figure 7B*). Inactivation of CDK12 is known to preferentially affect the expression of genes involved in DNA damage response (DDR) (*Blazek et al., 2011*; *Johnson et al., 2016*). We used RT-qPCR to monitor the mRNA levels of *BRCA1*, *BRCA2*, *ATR*, and *ERCC4*, four DDR genes known to be downregulated by CDK12 inhibition. HQ461 treatment resulted in reductions in the mRNA levels of all four tested DDR genes (*Figure 7C*). In cells with CDK12 G731E or G731R mutation, the inhibitory effect of HQ461 on POLII CTD Serine 2 phosphorylation and the expression of DDR genes were abolished (*Figure 7B and C*).

## Discussion

Based on our biochemical and biophysical data, we propose a model in which HQ461 binds to CDK12's ATP-binding pocket to generate an altered surface on CDK12 to interact with the BPC domain of DDB1 (*Figure 7D*). This model is supported by multiple lines of evidence. First, DSF experiments revealed direct binding of HQ461 to CDK12/CCNK instead of DDB1 (*Figure 5C and D*). Second, known inhibitors of CDK12 that occupy CDK12's ATP-binding pocket impeded HQ461's activity in inducing CDK12-DDB1 interaction and CCNK degradation (*Figure 5A* and *Figure 5—figure supplement 2C*). Third, CXMS analysis mapped the HQ461-induced protein-protein interaction interface to be between CDK12's kinase domain and DDB1's BPC domain (*Figure 5B*). Interestingly, the hot spot glycine 731 for HQ461-resistant mutations is located in the vicinity of this interface. G731E or G731R substitution did not affect HQ461 binding to CDK12 but impaired DDB1 recruitment (*Figure 5A and E*, *Figure 5—figure supplement 1A*). Replacing glycine (a neutral residue without a side chain) with glutamate or arginine (charged residues with bulky side chains) may cause steric hindrance and/or disruption of hydrophobic stacking to destabilize the CDK12-DDB1 interaction interface. High-resolution structural analyses of a CDK12-DDB1-HQ461 complex is needed to completely resolve the basis for HQ461's molecular-glue activity.

Previously known molecular-glue degraders all interact with a substrate-specific receptor of Cullin-RING ligases (CRL) to exert their functions. Auxin, thalidomide, and indisulam bind to TIR1, cereblon, and DCAF15, respectively (*Han et al., 2017*; *Ito et al., 2010*; *Tan et al., 2007*). HQ461 engages the CUL4-RING ligases (CRL4) in an unprecedented manner by directly interacting with the adaptor protein DDB1 of CRL4. CRLs catalyze polyubiquitination of their protein substrates by

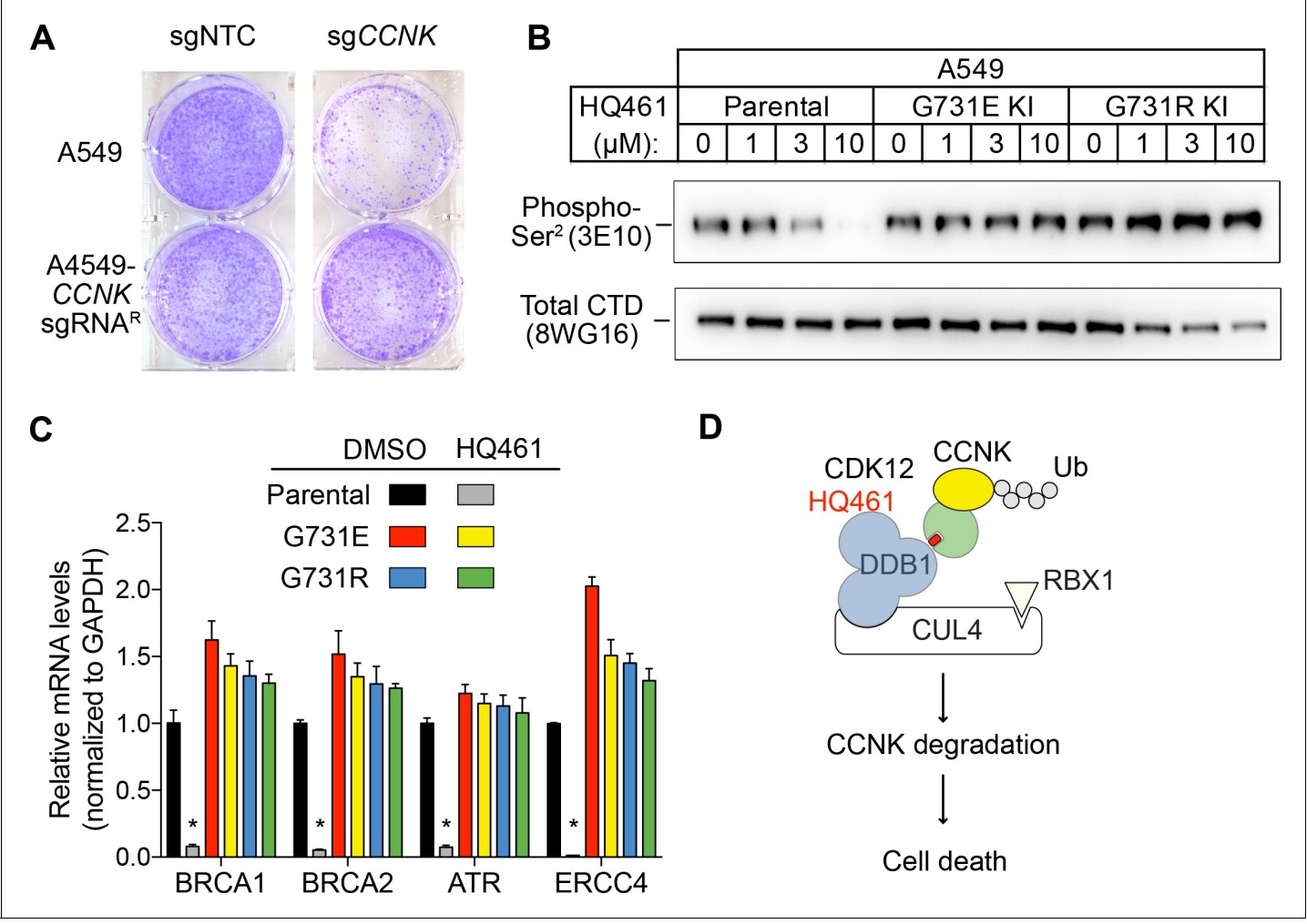

**Figure 7.** HQ461-mediated CCNK degradation causes cell death and impairs CDK12 function. (A) Depletion of CCNK by CRISPR reduces colony formation in parental A549 cells but not in A549 cells expressing an sgRNA-resistant *CCNK* cDNA. (B) Western blotting to examine levels of POLII CTD Serine 2 phosphorylation and total POLII CTD in parental A549 cells or A549 *CDK12* G731E or G731R knock-in cells following HQ461 treatment. (C) RT-qPCR to examine levels of BRCA1, BRCA2, ATR, and ERCC4 mRNAs in parental A549 cells or A549 *CDK12* G731E or G731R knock-in cells following 10 μM HQ461 treatment for 16 hr. Error bars represent standard deviations (SD) from three technical replicates. * represents p<0.05 (T-test, two tail, paired). (D) A model of HQ461's mechanism of action.

positioning them in a reactive zone. CDK12 appears to be outside this reactive zone, because HQ461 did not cause CDK12 ubiquitination in vitro (*Figure 4D* and *Figure 4—figure supplement 1*). In contrast, CDK12's partner protein CCNK seems to be optimally positioned for polyubiquitination. These findings suggest that HQ461-bound CDK12 is functionally equivalent to a substrate-specific receptor protein for CRL4. Interestingly, several viruses have evolved a similar strategy to use viral proteins as facultative substrate-specific receptors for DDB1 to degrade host restriction factors (*Li et al., 2006*; *Li et al., 2010*). In vivo, depletion of CCNK resulted in a modest reduction of CDK12 protein level (*Figure 3—figure supplement 1B*). The underlying mechanism is not understood at the moment, but one possibility is that CDK12 in the absence of CCNK is recognized and degraded by a quality control E3 ubiquitin ligase.

While we are preparing our work for publication, two studies identified CDK12 inhibitors that induce CDK12-DDB1 interactions to trigger CCNK degradation (*Mayor-Ruiz et al., 2020*; *Słabicki et al., 2020*). Slabicki et al. used a correlation of DDB1 mRNA level with drug-sensitivity across hundreds of cancer cell lines as a strategy to discover that the CDK12 inhibitor CR8 recruits CDK12 to DDB1 and leads to the degradation of CCNK. Mayor-Ruiz et al. examined 2000 cytotoxic compounds between normal and CRL-inactivated cells to identify several compounds whose

cytotoxicity requires CRL activity. We discovered HQ461 as a chemical toxin from a phenotype-based high-throughput screening project and primarily replied on chemical genetics (hypermutation in HCT-116 and CRISPR-Cas9 screening) to reveal HQ461's target. All three studies arrived at the same mechanism of action with distinct chemical scaffolds, suggesting that the CDK12-DDB1 interface is a hotspot for molecular glues.

CDK12 and CDK13 are two structurally similar kinases that has the ability to phosphorylate serine 2 of the POLII CTD heptad repeats (*Bösken et al., 2014*; *Dixon-Clarke et al., 2015*; *Greifenberg et al., 2016*). To execute their functions, CDK12 and CDK13 form two separate complexes with CCNK (*Blazek et al., 2011*; *Cheng et al., 2012*). Consistent with this notion, HQ461-mediated depletion of Cyclin K phenotypically mimics dual inhibition of CDK12 and CDK13 by the CDK12/13-selective inhibitor THZ531 to trigger cell death (*Zhang et al., 2016*). CDK12 inactivation is known to cause reduced POLII elongation and premature polyadenylation of select genes enriched in the DNA damage response pathway, resulting in a BRCAness phenotype (*Blazek et al., 2011*; *Dubbury et al., 2018*; *Johnson et al., 2016*). Several studies have shown that inhibiting CDK12 can sensitize cancer cells to PARP inhibitors (*Bajrami et al., 2014*; *Iniguez et al., 2018*; *Johnson et al., 2016*). Degradation of CCNK mediated by HQ461 also reduced the expression of DNA damage response genes (*Figure 7C*). It will be interesting to test whether HQ461, by depleting CCNK, also enhances the therapeutic efficacy of PARP inhibitors for cancer treatment.

# Materials and methods

**Key resources table**

| Reagent type (species) or resource | Designation | Source or reference | Identifiers | Additional information |
|---|---|---|---|---|
| Cell line (*Homo sapiens*) | A549 | Dr. Deepak Nijhawan's lab at University of Texas Southwestern Medical Center | | Male |
| Cell line (*Homo sapiens*) | HCT116 | Dr. Deepak Nijhawan's lab at University of Texas Southwestern Medical Center | | Male |
| Cell line (*Homo sapiens*) | HEK293T | Dr. Deepak Nijhawan's lab at University of Texas Southwestern Medical Center | | Female |
| Cell line (Spodoptera frugiperda) | SF9 | Dr. Sanduo Zheng from National Institute of Biological Sciences, Beijing | | |
| Cell line (Trichoplusia ni) | High Five | Dr. Sanduo Zheng from National Institute of Biological Sciences, Beijing | | |
| Cell line (*Homo sapiens*) | Free style 293 F | Dr. Linfeng Sun at China University of Science and Technology | | Female |
| Antibody | Anti-NRF2 (Rabbit monoclonal) | Abcam (ab62352) | RRID:AB_944418 | WB (1:4000) |
| Antibody | Anti-β-Actin-HRP (Mouse monoclonal) | Huaxingbio (HX18271) | | WB (1:10000) |
| Antibody | Anti-α-Tubulin-HRP (Rabbit polyclonal) | MBL Life Science (PM054-7) | RRID:AB_10695326 | WB (1:10000) |
| Antibody | Anti-DDB1 (Rabbit monoclonal) | Abcam (ab109027) | RRID:AB_10859111 | WB (1:10000) |
| Antibody | Anti-RBX1 (Rabbit polyclonal) | Proteintech (14895–1-AP) | RRID:AB_2179719 | WB (1:5000) |

*Continued on next page*

*Continued*

| Reagent type (species) or resource | Designation | Source or reference | Identifiers | Additional information |
|---|---|---|---|---|
| Antibody | Anti-UBE2G1 (Rabbit polyclonal) | Proteintech (12012–1-AP) | RRID:AB_10665812 | WB (1:2000) |
| Antibody | Anti-CDK12 (Rabbit polyclonal) | Cell Signaling Technology (11973S) | RRID:AB_2715688 | WB (1:4000) |
| Antibody | Anti-CCNK (Rabbit polyclonal) | Bethyl lab (A301-939A-T) | RRID:AB_2780226 | WB (1:4000) |
| Antibody | Anti-FLAG-HRP (Mouse monoclonal) | Sigma-Aldrich (A8592) | RRID:AB_439702 | WB (1:10000) |
| Antibody | Anti-HA (Mouse monoclonal) | BioLegend (901533) | RRID:AB_2801249 | WB (1:4000) |
| Antibody | Anti-RNA polymerase II subunit B1 (phospho CTD Ser-2) (Rat monoclonal) | Sigma-Aldrich (041571) | RRID:AB_10627998 | WB (1:5000) |
| Antibody | Anti-RNA polymerase II CTD repeat YSPTSPS (Mouse monoclonal) | Abcam (ab817) | RRID:AB_306327 | WB (1:5000) |
| Antibody | Anti-rabbit IgG (Goat polyclonal) | Cell Signaling Technology (7074S) | RRID:AB_2099233 | WB (1:10000) |
| Antibody | Anti-mouse IgG (Goat polyclonal) | Zsbio (ZB-2305) | RRID:AB_2747415 | WB (1:10000) |
| Antibody | Anti-rat IgG (Goat polyclonal) | Sino Biological (SSA005) | | WB (1:10000) |
| Recombinant DNA reagent | Lenti-EF1$\alpha$−3xFLAG -CDK12-P2A- BSD WT/GE/GR | This paper | | Described in Materials and methods; available upon request |
| Recombinant DNA reagent | pCDNA3.1-1XFLAG- CDK12 WT/GE/GR | This paper | | Described in Materials and methods; available upon request |
| Recombinant DNA reagent | pCDNA3.1-1XFLAG- CDK12-kinase domain (715–1052) WT/GE/GR | This paper | | Described in Materials and methods; available upon request |
| Recombinant DNA reagent | pCDNA3.1-3XFLAG- CDK12/13-kinase domain (715–1052/693-1030) WT/GE/GR | This paper | | Described in Materials and methods; available upon request |
| Recombinant DNA reagent | pcDNA3.1-CCNK-3xHA | This paper | | Described in Materials and methods; available upon request |
| Recombinant DNA reagent | pET28a-6xHis-Ub | This paper | | Described in Materials and methods; available upon request |
| Recombinant DNA reagent | pCMV-8×His-Ub | Dr. William Kaelin at Dana Farber Cancer Institute of Harvard | | |
| Recombinant DNA reagent | pPB-CAG-1xFLAG-UBA1 | This paper | | Described in Materials and methods; available upon request |
| Recombinant DNA reagent | pPB-CAG-1xFLAG-UBE2D3 | This paper | | Described in Materials and methods; available upon request |
| Recombinant DNA reagent | pPB-CAG-1xFLAG-CUL4A | This paper | | Described in Materials and methods; available upon request |

*Continued on next page*

*Continued*

| Reagent type (species) or resource | Designation | Source or reference | Identifiers | Additional information |
|---|---|---|---|---|
| Recombinant DNA reagent | pPB-CAG-RBX1 | This paper | | Described in Materials and methods; available upon request |
| Recombinant DNA reagent | pPB-CAG-1xFLAG-Avi-DDB1 | This paper | | Described in Materials and methods; available upon request |
| Recombinant DNA reagent | pFastBac-Strep-8×His-DDB1ΔBPB | This paper | | Described in Materials and methods; available upon request |
| Recombinant DNA reagent | pFastBac-6×His-CDK12KD | This paper | | Described in Materials and methods; available upon request |
| Recombinant DNA reagent | pFastBac-6×His-CCNKΔC | This paper | | Described in Materials and methods; available upon request |
| Recombinant DNA reagent | pFastBac-CAK1 | This paper | | Described in Materials and methods; available upon request |
| Recombinant DNA reagent | pCDNA3.1-CCNK-luc | This paper | | Described in Materials and methods; available upon request |
| Peptide, recombinant protein | HBx peptide | ChinaPetides | | ILPKVLHKRTLGLS |
| Commercial assay or kit | CellTiter-Glo | Promega | G7570 | |
| Commercial assay or kit | AlphaScreen Histidine (Nickel Chelate) Detection Kit | Perkin Elmer | 6760619C | |
| Commercial assay or kit | AlphaScreen Streptavidin Donor beads | Perkin Elmer | 6760619C | |
| Chemical compound, drug | Bortezomib | Targetmol | T2399 | CAS No. 179324-69-7 |
| Chemical compound, drug | THZ531 | Targetmol | T4293 | CAS No. 1702809-17-3 |
| Chemical compound, drug | Dinaciclib | Targetmol | T1912 | CAS No. 779353-01-4 |
| Chemical compound, drug | MLN4924 | Selleckchem | S7109 | CAS No. 905579-51-3 |
| Software, algorithm | BWA | *Li, 2013* | v0.7.17 | |
| Software, algorithm | samtools | *Li et al., 2009* | v1.10 | |
| Software, algorithm | GATK | *DePristo et al., 2011; McKenna et al., 2010; Van der Auwera et al., 2013* | v4.1.7 | |
| Software, algorithm | SnpSift | *Cingolani et al., 2012a* | v4.3t | |
| Software, algorithm | MAGeCK | *Li et al., 2014* | v0.5.9.2 | |
| Software, algorithm | ggplot2 | *Wickham, 2016* | | |
| Software, algorithm | GraphPad Prism | | v8.0.2 | |
| Software, algorithm | pLink 2 | *Chen et al., 2019* | | |

## Cell culture

The human cell lines A549, HCT-116, and HEK293T were obtained from Dr. Deepak Nijhawan's lab at University of Texas Southwestern Medical Center. Free style 293 F was a gift from Dr. Linfeng Sun at China University of Science and Technology. The insect cell lines Sf9 and High Five were gifts

from Dr. Sanduo Zheng from National Institute of Biological Sciences, Beijing. The identities for A549 and HCT-116 were confirmed by short tandem repeat (STR) analysis. All cell lines were confirmed to be mycoplasma free on a weekly basis using a PCR-based assay with primers: 5'-GGGAG-CAAACAGGATTAGATACCCT-3' and 5'-TGCACCATCTGTCACTCTGTTAACCTC-3'. Regular adherent cell culture methods were used to culture A549, HCT-116, and HEK293T cells in tissue-culture incubators with 5% $CO_2$ at 37°C. A549 were grown in RPMI-1640 medium with 10% fetal bovine serum (FBS) and 2 mM L-glutamine. HCT-116 and HEK293T cells were grown in DMEM medium with 10% FBS and 2 mM L-glutamine. Regular suspension cell culture methods were used to grow 293 F, Sf9, and high five cells. Free style 293 F cells were grown in SMM 293-TII expression medium (Sino Biological, Beijing, China) in a shaker incubator at 150 rpm, 37°C, 5% $CO_2$. Sf9 and High Five cells were cultured in ESF 921 medium (Expression Systems, Davis, USA) in a shaker at 140 rpm, 27°C.

## Chemicals

Bortezomib (CAS No. 179324-69-7), THZ531 (CAS No. 1702809-17-3), and dinaciclib (CAS No. 779353-01-4) were purchased from Targetmol (Topscience, Shanghai, China). MLN4924 (CAS No. 905579-51-3) was purchased from Selleckchem (Houston, USA). All of these chemicals were prepared as 10 mM stocks in DMSO (CAS: 67-68-5) purchased from Sigma-Aldrich, MO, USA and further diluted in DMSO to the desirable concentrations. HBx peptides (ILPKVLHKRTLGLS) was synthesized by ChinaPetides (Suzhou, China) with a purity of 96%.

## Construction of ARE-*luc2P* and TK-*luc2P* reporter cell lines

A549 cells were transiently transfected with the plasmid pGL4.37 (Promega, Madison, USA) and selected with 500 µg/ml of hygromycin to obtain a stable cell line harboring the ARE-*luc2P* reporter. TK-*luc2P* was cloned into the pLVX-IRE-Puro backbone by replacing its CMV promoter and multiple cloning site with HSV-TK promoter fused to *luc2P*. The resulting pLVX-TK-*luc2P*-IRES-Puro plasmid was packaged into lentivirus to transduce A549 cells. Stable cell lines were obtained by selection of transduced cells with 2 µg/ml of puromycin.

## High-throughput small-molecule screening

A screening library with 65,790 small molecules (Life Chemicals, Niagara-on-the-Lake, Canada) were used in primary screening. The screening procedures were as follows. Seven hundred A549 ARE-l*uc2P* cells in 50 µL of medium were plated per well in 384-well flat clear bottom white polystyrene TC-treated microplates (Corning, Corning, USA) and allowed to attach to plates overnight. Compounds from the screening library were added to the cells with Biomek FXP automated workstation at a final concentration of 10 µM. Twelve hours later, ARE-*luc2P* activities were measured with Bright-Glo luciferase assay system (Promega, Madison, USA) following vendor instructions. Luminescence was measured on a EnVision multimode plate reader (PerkinElmer, Waltham, USA). Luminescence data on each plate was Z-score normalized using the equation $Z = \frac{x-\mu}{\sigma}$, where x is the luminescence value of a well, µ is the mean and $\sigma$ is the standard deviation of all values on one plate. Z scores from all the plates were aggregated and a cutoff (Z-score < -2.5) was used to select 515 hits from the primary screening. These hits were picked from the screening library and used for counter screening using both A549 ARE-*luc2P* and A549 TK-*luc2P* cells. Normalization to negative control (DMSO) was used to calculate a % of inhibition value for every compound on the two *luc2P* reporters. A ratio (% inhibition of TK-luc2P / % inhibition of ARE-luc2P) greater than 2.5 was used to identify compounds with ARE-*luc2P* selectivity.

## Cell viability assay

Seven hundred A549 cells or 1,000 HCT-116 cells in 50 µL of medium were plated per well in 384-well flat clear bottom white polystyrene TC-treated microplates (Corning, Corning, USA). After overnight attachment, cells were dosed with a serial dilution of HQ461 with a D300e digital dispenser (Tecan, Männedorf, Switzerland). Cell survival was measured 72 hr later using CellTiter-Glo luminescent cell viability assay kit (Promega, Madison, USA) following vendor instructions. Luminescence was recorded by EnVison multimode plate reader (PerkinElmer, Waltham, USA). IC50 was determined with GraphPad Prism using baseline correction (by normalizing to DMSO control), the asymmetric (five parameter) equation, and least squares fit.

## Pooled genome-wide CRISPR-Cas9 sgRNA screening of HQ461 resistance

The human CRISPR knockout pooled library (Brunello) (*Doench et al., 2016*) was a gift from Dr. Feiran Lu at University of Texas Southwestern Medical Center. The sgRNA library was packaged into lentiviral vector for delivery into A549 cells as described (*Joung et al., 2017*). The screening parameters were as follows. Thirty million A549 cells were infected at a multiplicity of infection (MOI) of ~0.3. Infected cells were passaged every 2 days in the presence of 10 µg/ml of puromycin for 1 week with a population size of at least 200 million. Afterwards, A549 cells transduced with the sgRNA library were passaged every 2 days in the presence of 4 µM HQ461 or the vehicle DMSO (0.1% v/v) for 3 weeks with a population size of at least 10 million. Two biological replicates were performed for both HQ461 and DMSO treatment.

## Isolation of genomic DNA

Cells were lysed in 10 mM Tris-HCl, pH 8.0, 100 mM EDTA, 0.5% SDS, 200 µg/ml RNase A and incubated at 37°C for 1 hr followed by the addition of 1 mg/ml of proteinase K for an overnight incubation at 50°C. The resulting lysates were extracted three times with phenol solution equilibrated with 10 mM Tris-HCl, pH 8.0, 1 mM EDTA (Sigma-Aldrich, MO, USA P4557) mixed with chloroform (1:1 v/v) using tubes with phase lock gel (Tiangen, Beijing, China). The aqueous phase was then mixed with 0.2 vol of 10 M ammonium acetate and 1 vol of isopropanol, resulting the immediate formation of cloudy DNA precipitates. Precipitated DNA was then transferred by a pipet tip to a tube containing 75% (v/v) ethanol. This process was repeated twice to ensure complete removal of residual organic solvents. Afterwards, genomic DNA was dissolved in 1 mM Tris-HCl, pH 8.0, 0.1 mM EDTA by incubation at 50°C for 3 hr.

## PCR amplification and next generation sequencing of sgRNAs

DNA fragments containing sgRNA sequences were amplified from isolated genomic DNA by two rounds of PCR using NEBNext Ultra II Q5 master mix (NEB, Ipswich, USA). For the first round of PCR, forty-eight 50 µL PCR reactions (each containing 2 µg of genomic DNA template) were performed with the forward primer NGS-Lib-KO-Fwd-1 (5'- CCTACACGACGCTCTTCCGATC TNNNNNNNNNNNNNNNNNNNNGCTTTATATATCTTGTGGAAAGGACGAAACACC −3') and the reverse primer NGS-Lib-KO-Rev-0 (5'-CAGACGTGTGCTCTTCCGATCTCCGACTCGGTGCCAC TTTTTCAA-3') with a thermal cycler program consisting of initial denaturation at 98°C for 5 min, followed by 16 cycles of (98°C denaturation for 10 s, 69°C annealing for 30 s, and 65°C extension for 45 s), and a final extension at 65°C for 5 min. Products of the first-round PCR were pooled and purified by a DNA clean and concentrator kit (Zymo, Irvine, USA) and diluted to 2 ng/µL. For the second round of PCR, six 50 µL PCR reactions (each containing 2 ng of the purified first-round PCR product) were performed for each sample with the forward primer NGS-Lib-KO-Fwd-2 (5'-AATGATACGGC-GACCACCGAGATCTACACTCTTTCCCTACACGACGCTCTTCCGATCT-3') and one of four indexed reverse primers (NGS-Lib-KO-Rev-1, 2, 3, 4) (5'-CAAGCAGAAGACGGCATACGAGAT-8-nucleotide index-GTGACTGGAGTTCAGACGTGTGCTCTTCCGATCT-3'). The same cycling conditions were used for second-round PCR as the first-round PCR except 18 cycles were used. Products of the second-round PCR reactions were subjected to electrophoresis on a 2% agarose gel. The expected ~300 bp amplicons were excised and extracted from the gel and sequenced by Illumina HiSeq PE150 (Novogene, Beijing, China). Raw sequencing data have been deposited to NCBI GEO (GSE153700).

## Data analysis for genome-wide CRISPR knockout screening

CRISPR screening data were analyzed by MAGeCK (Model-based Analysis of Genome-wide CRISPR-Cas9 Knockout, v0.5.9.2) (*Li et al., 2014*) to discover candidate genes by comparing experimental condition (HQ461) with control condition (DMSO). Raw Fastq files were directly loaded into MAGeCK and the 'count' command was used to collect read counts from Fastq files and to generate the sgRNA read count table. Next, the 'test' command was used to perform statistical test from the count table, outputting $\log_2$ fold change, p value, and false discovery rate (FDR). The result table generated by MAGeCK was loaded into ggplot2 (*Wickham, 2016*) in R to generate a volcano plot.

## Validation of top-ranking genes from CRISPR-Cas9 knockout screening

Two sgRNAs per gene were chosen from the Brunello library for DDB1 (5'-CATTGTCGATATGTGCG TGG-3' and 5'-CTACCAACCTGCGATCACCA-3'), RBX1 (5'- AGTACACTCTTCTGAAGTAG-3' and 5'-ATGGATGTGGATACCCCGAG-3'), and UBE2G1 (5'-ACTTACTAAAGTGTATCTGG-3' and 5'- A TGAAAAGCCAGAGGAACGC-3'). Annealed sgRNA oligos were cloned into the lentiCRISPR v2 vector. Lentiviral packaging was performed by co-transfecting the resulting plasmids with psPAX2 (Addgene 12260) and pMD2.G (Addgene 12259) into 293 T cells. Media collected from transfected 293 T cells was used to infect A549 cells at MOI ~ 5 for 4 days. The resulting cells were tested for their sensitivity to HQ461 using methods describe in the previous session cell viability assay.

## Isolation of HQ461 resistant HCT-116 clones for pooled whole exome-sequencing

Parental HCT-116 were plated sparsely on 10 cm plates to allow the isolation of ten individual clones. Each of these ten clones were expanded to one confluent 15 cm plate to establish ten independent cell populations. One million cells from each population were then plated on one 10 cm plate and treated with 20 μM HQ461 continuously for two weeks, with media change every 3 to 4 days. HQ461 resistant clones emerged from 5 out of the 10 populations. These clones were isolated, expanded, and tested for their sensitivity to HQ461 using cell viability assay. Five HQ461-resistant clones selected from independent populations were subjected to genomic DNA isolation as described in the previous session. Their genomic DNAs were then pooled in equal amounts for whole-exome sequencing at 200x average coverage (Novogene, Beijing, China). As a control, genomic DNAs isolated from the corresponding cell populations before HQ461 selection were also pooled for whole-exome sequencing. Raw sequencing data have been deposited to NCBI GEO (GSE153707).

## Data analysis for whole-exome sequencing

Fastq files were aligned to the human reference genome version GRCh38 (hg38) using Burrows-Wheeler Aligner (BWA, v0.7.17) (*Li, 2013*) to generate sam files, which were then transformed to bam files through samtools (v1.10) (*Li et al., 2009*). Next, bam files were passed to the Genome Analysis Toolkit (GATK, v4.1.7) (*DePristo et al., 2011*; *McKenna et al., 2010*; *Van der Auwera et al., 2013*) to call SNVs, deletions, and insertions. To call variants, several tools of GATK were utilized, including BaseRecalibrator to adjust base quality scores in the bam files using known variants in dbSNP database (build 146) and 1000 Genomes database, Mutect2 for variant calling, GetPileupSummaries and CalculateContamination to estimate and remove cross-sample contamination, CollectSequencingArtifactMetrics and FilterByOrientationBias to exclude sequence context-dependent artifacts, and FilterMutectCalls to filter out variants from some common sources of error. Finally, variants generated by GATK were annotated using snpEff (v4.5) (*Cingolani et al., 2012b*) and SnpSift (v4.3t) (*Cingolani et al., 2012a*).

## Sanger sequencing of *CDK12*

Genomic DNAs isolated from HQ461-resistant clones were used as template for amplification of a DNA sequence flanking glycine 731 of CDK12 with primers 5'- GTAAAACGACGGCCAGTGTTGTCC TCGTTATGGAGAAAGAA-3' and 5'-CTGTGTCTTTGTCCTTGGCTTTAT-3'. PCR amplicons were Sanger sequenced with the M13F sequencing primer: 5'- GTAAAACGACGGCCAGT-3'.

## CRISPR-Cas9 knock-in of *CDK12* G731E or G731R alleles

An sgRNA targeting *CDK12* (5'-GTGGACAAGTTTGACATTAT-3') was cloned into the pSpCas9(BB)—2A-eGFP (PX458) vector (Addgene 48138). For CDK12 G731E/R knock-in, 1 million A549 cells were nucleofected (using 4D-Nucleofector, Lonza, Basel, Switzerland) with PX458-sg*CDK12* and single-stranded oligodeoxynucleotides (G731E: 5'-ttatatacttggccataggttccttctccaataatTTCaataatgt-caaacttgtccacacagcgtttcccccagtc-3'; G731R: 5'-ttatatacttggccataggttccttctccaataatGCGaataatgt-caaacttgtccacacagcgtttcccccagtc-3'). Afterwards, cells were exposed to 5 μM 0461 for 2 weeks to select for cells with *CDK12* G731E/R knock-in. Cells that survived 0461 treatment were recovered and confirmed to have the correct G731E/R genomic conversion via Sanger sequencing.

## Generation of stable cell lines expressing 3xFLAG-CDK12

An ORF encoding *CDK12* isoform 2 (NM_015083.3) was PCR amplified from A549 cDNA and cloned into a pCDNA3.1-N-3xFLAG vector which was derived from the pCDNA3.1 vector. G731E and G731R mutations in *CDK12* were introduced by overlap extension PCR. After sequencing verification, wild-type and mutant 3xFLAG-*CDK12* sequences were subcloned into lentiCas9-Blast (Addgene 52962) by replacing the Cas9 ORF with 3xFLAG-*CDK12*. The resulting lenti-EF1α−3xFLAG-CDK12-P2A-BSD WT/GE/GR constructs were packaged into lentiviral vectors to transduce A549 cells. Stable cell lines were obtained by selection with 20 μg/ml of blasticidin.

## Antibodies for western blotting

Standard SDS-PAGE and western blotting procedures were used with the following modifications. For preparation of total lysates, cells were rinsed with DPBS to remove residual medium and then lysed in 20 mM HEPES-NaOH, pH 8.0, 10 mM NaCl, 2 mM MgCl$_2$, 1% SDS freshly supplemented with 0.5 units/μL of benzonase and 1x cOmplete, Mini, EDTA-free protease inhibitor cocktail (Roche, Bazel, Switzerland). Protein concentrations of the resulting lysates were quantified by the BCA method. Between 30 to 60 μg of proteins were resolved on SDS-PAGE and transferred to nitrocellulose membranes with a pore size of 0.5 μm. Membranes were blocked in 5% nonfat milk PBST (0.1% v/v Tween-20) for 30 min before blotting with antibodies. The following primary antibodies were used by dilution in 5% nonfat milk PBST: anti-NRF2 (Abcam, Cambridge, UK, ab62352, 1:4,000), anti-β-Actin-HRP (Huaxingbio, Beijing, China, HX18271,1:10,000), anti-α-Tubulin-HRFP (MBL Life Science, Japan, PM054-7, 1:10,000), anti-DDB1 (Abcam, Cambridge, UK, ab109027, 1:10,000), anti-RBX1 (Proteintech, Wuhan, China, 14895–1-AP, 1:5,000), anti-UBE2G1 (Proteintech, Wuhuan, China, 12012–1-AP, 1:2,000), anti-CDK12 (Cell Signaling Technology, Danvers, USA, 11973S,1:4,000), anti-CCNK (Bethyl lab, Montgomery, USA, A301-939A-T,1:4,000), anti-FLAG-HRP (Sigma-Aldrich, MO, USA, A8592, 1:10,000), anti-HA (BioLegend, San Diego, USA, 901533, 1:4,000), anti-RNA polymerase II subunit B1 (phospho CTD Ser-2) clone 3E10 (Sigma-Aldrich, MO, USA, 04–1571, 1:5,000), and anti-RNA polymerase II CTD repeat YSPTSPS antibody clone 8WG16 (Abcam, Cambridge, UK, ab817, 1:5,000). The following HRP-linked secondary antibodies were used by dilution in PBST: anti-rabbit IgG (Cell Signaling Technology, Danvers, USA, 7074S, 1:10,000), anti-mouse IgG (Zsbio, Beijing, China, ZB-2305, 1:10,000), and anti-rat IgG, (Sino Biological, Beijing, China, SSA005, 1:10,000). M5 HiPer ECL Western HRP Substrate (Mei5bio, Beijing, China, MF074-01) was used for the detection of HRP enzymatic activity. Western blot images were taken with a VILBER FUSION FX7 imager.

## In vivo polyubiquitination of CCNK

Wild-type and G731E/R ORFs of 3 × FLAG-*CDK12*, either full length or the kinase domain (from resides 715–1052), were cloned into the pCDNA3.1-N-3xFLAG vector. An ORF encoding CCNK (NM_001099402.2) was PCR amplified with A549 cDNA and cloned into a pCDNA3.1-C-3xHA vector. pCMV-8 ×His Ub was a gift from Dr. William Kaelin at Dana Farber Cancer Institute of Harvard. To set up cells for the in vivo ubiquitination assay, 0.6 million HEK293T cells were seeded per well in six-well tissue-culture plates. After overnight attachment, these cells were transfected with 500 ng of pCMV 8 × His Ub, 25 ng of pCDNA3.1-CCNK-3xHA, and 50 ng of pCDNA3.1-3xFLAG-CDK12. Two days later, cells were pretreated with 1 μM of bortezomib for 2 hr, followed by treatment with DMSO or three doses of HQ461 (1, 3, 10 μM) for 4 hr. Cells from each well were rinsed once with DPBS and lysed in 500 μL of buffer 1 (25 mM Tris-HCl, pH 8.0, 8 M urea, 10 mM imidazole) freshly supplemented with one units/μL of benzonase and 1x cOmplete, Mini, EDTA-free protease inhibitor cocktail (Roche, Bazel, Switzerland). Lysates were clarified by spinning at 15,000 xg at 4°C for 10 min. Clarified lysate of each sample was mixed with 10 μL of Ni NTA magarose beads (SMART Life Sciences, Changzhou, China, SM00801) prewashed with lysis buffer. Tubes containing the bead-lysate mixtures were rotated for 4 hr at room temperature followed by two washes with buffer 1, one wash with a 1:3 mixture of buffer 1: buffer 2 (25 mM Tris-HCl, pH 6.8, 20 mM imidazole), and one wash with buffer 2. His-Ub-conjugated proteins were eluted from the beads by boiling in 50 μL of 1x SDS sample buffer supplemented with 300 mM imidazole.

## Co-expression of 3xFLAG-CDK12 or 3xFLAG-CDK13 kinase domain with CCNK-3xHA

ORFs encoding residues 715–1052 of CDK12 and residues 693–1030 of CDK13 were cloned into a pCDNA3.1-N-3xFLAG vector. For setting up cells for co-transfection, 0.6 million 293 T cells were seeded per well in 6-well tissue-culture plates and allowed to attach to plates overnight. These cells were then transfected with 50 ng of pCDNA3.1-3xFLAG-CDK12/13 kinase domain, 50 ng of pCDNA3.1-CCNK-3xHA, and 400 ng of the empty pCDNA3.1 vector. Twenty-four hours post transfection, cells were treated with HQ461 for 8 hr, and lysates were collected for western blotting with anti-FLAG and anti-HA antibodies.

## N-terminal 3xFLAG-tagging of endogenous *CDK12*

An sgRNA sequence (5′-GCCCAATTCAGAGAGACATG-3′) targeting the genomic region immediately downstream the CDK12 start codon was cloned into the PX458 vector (Addgene 48138). The 3xFLAG knock-in repair template was constructed in a pTOPO-TA vector (Mei5bio, Beijing, China) containg a BSD-P2A-3xFLAG sequence flanked by two 500 bp homology arms matching upstream and downstream sequences of the *CDK12* genomic locus. For endogenous tagging of *CDK12*, 1 million A549 cells were nucleofected (using 4D-Nucleofector, Lonza, Basel, Switzerland) with 1 μg of PX458-sg*CDK12* and 1 μg of the repair template. Selection with 30 μg/ml of blastistin was performed until clones appeared. Multiple clones were isolated and successful integration of N-terminal 3xFLAG tag was validated by western blotting with anti-FLAG-HRP (Sigma-Aldrich, MO, USA, A8592, 1:10,000).

## Co-immunoprecipitation of CCNK and DDB1 with 3xFLAG-CDK12 complex

Anti-FLAG M2 antibody (Sigma-Aldrich, MO, USA F3165) was coupled to magnetic epoxy beads (Beijing Yunci Technology Co., Beijing, China) at the ratio of 10 μg of anti-FLAG antibody/mg of beads in the presence of 1 M ammonium acetate and 0.1 M sodium phosphate, pH 7.4 at 37℃ overnight. A549 3xFLAG-*CDK12* knock-in cells were detached from plates by scraping, washed in DPBS, and then frozen in liquid nitrogen. Frozen cells were pulverized using a mixer mill MM 400 (Retsch, Haan, Germany) with two rounds of 1 min ball milling at 30 Hz. Per experiment, 25 mg of grinded cell powder was solubilized with 250 μL of IP buffer (50 mM HEPES, pH 7.4, 300 mM NaCl, 0.1% Tween-20) supplemented with 1x cOmplete, Mini, EDTA-free protease inhibitor cocktail (Roche, Bazel, Switzerland). The resulting lysates were centrifuged at 15,000 g for 10 min at 4℃. Clarified lysates were supplemented with HQ461 or DMSO and incubated at 4℃ for 30 min. Afterwards, 0.2 mg of anti-FLAG-conjugated magnetic beads were mixed with clarified lysates for 15 min on a rotating platform at 4℃, followed by three washes with IP buffer supplemented with HQ461 or DMSO. Bound proteins were eluted from magnetic beads with 1 mg/ml of 3 × FLAG peptide (Sigma-Aldrich, MO, USA F4799) with agitation at 4℃ for 30 min. Eluted proteins were subjected to SDS-PAGE and western blotting with anti-FLAG-HRP, anti-CCNK, and anti-DDB1. For CoIP of CDK12/CCNK with exogenously supplemented His-DDB1, A549 cells stably expressing 3xFLAG-CDK12 WT or G731E/R mutants were lysed in lysis buffer (50 mM HEPES, pH 7.4, 300 mM NaCl, 0.1% Triton-X100) followed by immunoprecipitation with anti-FLAG-conjugated magnetic beads. After three washes with binding buffer (50 mM HEPES, pH 7.4, 300 mM NaCl, 0.5% NP-40), beads were mixed with 500 nM recombinant His-DDB1 and HQ461 in binding buffer and incubated at 4℃ for 30 min. After three washes with binding buffer supplemented with HQ461, bound proteins were eluted with 1 mg/ml 3xFLAG peptide.

## Purification of ubiquitin and enzymes for in vitro ubiquitination assay

Human Ubiquitin ORF was cloned into pET28a vector with an N-terminal 6xHis tag. The resulting pET28a-6xHis-Ub plasmid was transformed into the *E. coli* strain BL21 (DE3) and grown at 37℃ until OD$_{600}$ reached ~0.8. His-Ub expression was induced overnight by 0.5 mM IPTG at 18℃. Cells were collected by centrifugation at 4000 g and resuspended with lysis buffer containing 50 mM Tris-HCl, pH 7.5, 500 mM NaCl, 20 mM imidazole, 5% glycerol. Cell lysis was performed by French Press. After 20,000 g centrifugation for 1 hr at 4℃, the supernatant was incubated with Ni-NTA resin. After extensive wash with lysis buffer, His-Ub was eluted from Ni-NTA resin with 50 mM Tris-HCl, pH 7.5,

500 mM NaCl, 300 mM imidazole, 5% glycerol. Fractions containing His-Ub were pooled and concentrated, followed by gel filtration on a Superdex 200 10/300 GL column (GE Healthcare, Chicago, USA) in gel filtration buffer (50 mM Tris-HCl, pH7.5, 10 mM NaCl, 1 mM DTT). Fractions containing His-Ub were pooled and concentrated by a 3 KDa MWCO Ultra centrifugal filter (Amicon), flash frozen with liquid nitrogen, and stored at −80°C before use.

Human ORFs encoding UBA1 (residues 49–1058, NP_003325.2), UBE2G1 (NP_003333.1), UBE2D3 (NP_003331.1), CUL4A (NP_001008895.1) were cloned into pPB-CAG-1xFLAG vector. ORF encoding human RBX1 (NP_055063.1) was cloned into pPB-CAG vector without a tag. Plasmids were transfected into 293 F cells at the condition of 0.1 mg of plasmid for 100 million 293 F cells at a density at 1 million cells per ml. FLAG-UBA1, FLAG-UBE2G1, FLAG-UBE2D3 were separately expressed, whereas FLAG-CUL4A was co-expressed with RBX1. Cells were collected 48 hr post transfection by centrifugation at 1000 g and then lysed in binding buffer (45 mM Tris-HCl, pH 7.8, 180 mM NaCl) supplemented with 0.2% Triton X-100, 1 mM DTT, and 1x cOmplete, Mini, EDTA-free protease inhibitor cocktail (Roche, Bazel, Switzerland). After 20,000 g centrifugation, the clarified lysates were mixed with anti-FLAG M2 affinity gel (Sigma-Aldrich, MO, USA, A2220) for 2 hr at 4°C on a rotator. After extensive washes with binding buffer plus 0.1% Triton X-100, FLAG-tagged proteins were eluted from beads by 0.2 mg/ml FLAG peptide (Sangon Biotech, Shanghai, China, T510060-0005). The protein-containing fractions were concentrated by ultracentrifugation and fractionated by an ENrich SEC650 gel filtration column (Bio-Rad, Hercules, USA) in binding buffer. Fractions containing target proteins were pooled, concentrated by ultrafiltration, flash frozen with liquid nitrogen and stored at −80°C.

Human DDB1(NP_001914.3) was cloned into pFastBac with an N-terminal Strep-tag and an 8 × His tag. Expression and purification of His-DDB1 from Sf9 cells followed identical procedures as described in *Purification of DDB1ΔBPB*.

## In vitro ubiquitination of CCNK

293 F cells were transiently transfected with pPB-CAG-1xFLAG-3xHA-CDK12 and pPB-CAG-CCNK-3xV5 plasmids. Forty-eight hours later, FLAG-3xHA-CDK12 in complex with CCNK-3xV5 was immunopurified from transfected cells using anti-FLAG antibody-conjugated magnetic beads and eluted with 0.2 mg/ml FLAG peptide. In vitro ubiquitination reactions were performed by mixing FLAG-3xHA-CDK12/CCNK-3xV5 with 0.2 μM UBA1, 0.5 μM UBE2G1, 0.5 μM UBE2D3, 0.8 μM CUL4A-RBX1, 1 μM DDB1, and 100 μM ubiquitin in a buffer containing 50 mM HEPES, pH 7.5, 5 mM MgCl$_2$, 5 mM ATP, 75 mM sodium citrate, and 0.1% Tween-20. HQ461 or DMSO were added to the reaction 30 min prior to the addition of enzymes. Reactions were incubated for 1 hr at 30°C with agitation and then quenched with SDS sample buffer, followed by SDS-PAGE and western blotting with anti-HA and anti-V5 antibodies.

## Expression and purification of CDK12KD/CCNKΔC

Human CDK12KD (residues 715–1052) and human CCNKΔC (residues 11–267) were cloned into the baculoviral transfer vector pFastBac with an N-terminal 6xhis tag followed by a PreScission Protease cleavage site. G731E and G731R mutations were introduced to CDK12KD by overlap extension PCR. ORF of Cdk-activating kinase (CAK1) from *Saccharomyces cerevisiae* (Uniprot P43568) was cloned into pFastBac without a tag. Bacmid DNA was prepared in the *E. coli* strain DH10Bac and used to generate baculovirus by transfection into Sf9 cells. Sf9 or High Five cells were coinfected with baculovirus for CDK12KD (WT or G731E/R mutants), CCNKΔC, and yeast CAK1. Cells were lysed by sonication in binding buffer (50 mM HEPES, pH 7.5, 500 mM NaCl, 5% glycerol) supplemented with 20 mM imidazole and 1x cOmplete, Mini, EDTA-free protease inhibitor cocktail (Roche, Bazel, Switzerland). The lysate was clarified by centrifugation at 15,000 g for 1 hr at 4°C. Recombinant proteins were purified by Ni NTA Beads (Smart Lifesciences, Changzhou, China, SA004010) with 30 mM imidazole for washing and 300 mM imidazole for elution. Fractions containing CDK12KD/CCNKΔC were treated with recombinant PreScission Protease at an enzyme to substrate ratio of 1:100 at 4°C overnight to remove the 6xHis tag. The cleaved proteins were buffer-exchanged using 30 KDa MWCO Amicon ultra centrifugal filters (Merck Millipore, Burlington, USA) to 50 mM MES, pH 6.5, 100 mM NaCl, 0.5 mM TCEP, and manually loaded onto a 5 mL HiTrap Q column (GE Healthcare, Chicago, USA). The flow-through containing CDK12KD/CCNKΔC was concentration by

ultrafiltration, loaded onto ENrich SEC650 size exclusion column (Bio-Rad, Hercules, USA), and then fractionated in 50 mM HEPES, pH 7.5, 300 mM NaCl, 0.5 mM TCEP. Fractions containing CDK12KD/CCNKΔC were pooled and concentrated by ultrafiltration, flash frozen with liquid nitrogen, and stored at −80°C. For the AlphaScreen assay, His-tag removal was omitted from the above procedures.

## Purification of DDB1ΔBPB

Human DDB1ΔBPB (removing residues 400–704) was cloned into pFastBac with an N-terminal Strep-tag and an 8 × His tag. Sf9 cells infected with DDB1ΔBPB were lysed by sonication in binding buffer (50 mM Tris-HCl, pH 7.5, 500 mM NaCl, 10% glycerol, 2 mM TCEP, 1 mM PMSF, 10 mM imidazole) supplemented with 1x cOmplete, Mini, EDTA-free protease inhibitor cocktail (Roche, Bazel, Switzerland). DDB1ΔBPB was purified from clarified lysates by Ni NTA Beads and eluted in 50 mM Tris-HCl, pH 7.5, 500 mM NaCl, 10% glycerol, 2 mM TCEP, 1 mM PMSF, 300 mM imidazole. The fractions containing DDB1ΔBPB were pooled, concentrated, and then subjected to gel filtration on an ENrich SEC650 size exclusion column (Bio-Rad, Hercules, USA in 50 mM HEPES, pH 7.5, 300 mM NaCl, 0.5 mM TCEP).

## Purification of biotinylated FLAG-Avi-DDB1

Full-length human DDB1 was cloned into a pPB-CAG vector with an N-terminal FLAG-tag (for purification) and an Avi-tag (for biotinylation). The resulting pPB-CAG-FLAG-Avi-DDB1 construct was co-transfected with pPB-BirA (1:1 mass ratio) into 293 F cells at the condition of 0.1 mg of plasmid for 100 million 293 F cells at a density at 1 million cells per ml. D-Biotin (Targetmol, T1116) was added to the transfected cells at a final concentration of 50 µM. Cells were collected 48 hr post-transfection by 1,000 g centrifugation and resuspended in binding buffer (50 mM Tris-HCl, pH 7.5, 500 mM NaCl, 10% glycerol, 2 mM TCEP, 1 mM PMSF) supplemented with 1x cOmplete, Mini, EDTA-free protease inhibitor cocktail (Roche, Bazel, Switzerland). After sonication, lysates were clarified by centrifugation at 15,000 g for 1 hr at 4°C. The supernatant containing recombinant proteins were incubated with anti-FLAG M2 affinity gel (Sigma-Aldrich, MO, USA, A2220) for 2 hr on a rotator at 4°C. After extensive wash with binding buffer, FLAG-Avi-DDB1 was eluted from beads by 0.2 mg/ml FLAG peptide (Sangon Biotech, Shanghai, China, T510060-0005). The protein-containing fractions were concentrated by ultracentrifugation and further purified by gel filtration on an ENrich SEC650 size exclusion column (Bio-Rad, Hercules, USA).

## FLAG-DDB1:CDK12KD/CCNKΔC pull down assay

FLAG-Avi-DDB1 (18.2 µg) was mixed with CDK12KD/CCNKΔC (10 µg) at 1:1 molar ratio in 500 µL of pulldown assay buffer containing 50 mM HEPES, pH 7.5, 300 mM NaCl, 0.5 mM TCEP. The protein mix was then supplemented with DMSO or HQ461. After 30 min of incubation on ice, the protein-compound mix was incubated with anti-FLAG M2 affinity gel for another 30 min on a rotator at 4 °C. Protein-bound beads were washed three times with the pulldown assay buffer and bound proteins were eluted with 0.2 mg/mL FLAG peptide. Eluted proteins were analyzed by SDS-PAGE followed by coomassie blue staining.

## AlphaScreen assay

Biotinylated FLAG-Avi-DDB1 (3 µM) and 6xHis-CDK12KD/6xHis-CCNKΔC (WT or G731E/R mutants) (3 µM) were mixed in an assay buffer (50 mM HEPES, pH7.5, 300 mM NaCl, 0.5 mM TCEP, 0.1% NP-40, 0.1% BSA) and a serial dilution of HQ461.The protein-compound mix was transferred to 384-well plates (Corning, Corning, USA); 15 µL per well. After a 1 hr room temperature incubation, 10 µL of nickel chelate (Ni-NTA) acceptor beads (PerkinElmer, Waltham, USA, Waltham, USA, 6760619C) (1:100 diluted with assay buffer) was added per well. After incubation in the dark for 1 hr at room temperature, 10 µL of streptavidin donor beads (PerkinElmer, Waltham, USA, 6760619C) (1:100 diluted in assay buffer) was added per well. After another 1 hr incubation, AlphaScreen signal was measured on the EnVision multimode plate reader (PerkinElmer, Waltham, USA). For THZ531 competition, FLAG-Avi-DDB1, 6xHis-CDK12KD/6xHis-CCNKΔC mix was preincubated with 5 µM of THZ531 for 30 min on ice followed by the procedures described above.

## Chemical cross-linking mass spectrometry

A mixture of 23.4 μg of FLAG-Avi-DDB1 and 12.2 μg of CDK12KD/CCNKΔC (1:1 molar ratio) was incubated in the presence of 10 μM HQ461 for 30 min on ice, followed by cross-linking with 1 mM DSS (Thermo Fisher, Waltham, USA) or 1 mM BS3 (Thermo Fisher, Waltham, USA). The cross-linking reactions were incubated for 1 hr on a horizontal rotator at room temperature, followed by quenching with 20 mM ammonium bicarbonate for 20 min. Cross-linked samples were precipitated by 25% trichloroacetic acid (final concentration) for 30 min on ice. The resulting pellets were washed twice using cold acetone, air-dried and then dissolved, assisted by sonication, in 8 M urea, 20 mM methylamine, 100 mM Tris-HCl, pH 8.5. After reduction (5 mM TCEP, room temperature, 20 min) and alkylation (10 mM iodoacetamide, room temperature, 15 min in the dark), the samples were diluted to 2 M urea with 100 mM Tris-HCl, pH 8.5. Denatured proteins were digested by trypsin at 1/50 (w/w) enzyme/substrate ratio at 37 °C overnight and the reactions were quenched with 5% formic acid (final concentration).

## Mass spectrometry analysis of cross-linked DDB1/CDK12KD/CCNKΔC

The LC-MS/MS analysis was performed on an Easy-nLC 1000 HPLC (Thermo Fisher, Waltham, USA) coupled to a Q-Exactive HF mass spectrometer (Thermo Fisher, Waltham, USA). Peptides were loaded on a pre-column (75 μm ID, 4 cm long, packed with ODS-AQ 12 nm-10 μm beads) and separated on an analytical column (75 μm ID, 12 cm long, packed with Luna C18 1.9 μm 100 Å resin) with a 60 min or 120 min linear gradient at a flow rate of 250 nl/min. The 60 min linear gradient was as follows: 0–5% B in 1 min, 5–30% B in 45 min, 30–100% in 4 min, 100% for 10 min (A = 0.1% FA, B = 100% ACN, 0.1% FA); the 120 min linear gradient was as follows: 0–5% B in 1 min, 5–30% B in 100 min, 30–100% in 4 min, 100% for 15 min (A = 0.1% FA, B = 100% ACN, 0.1% FA). The top fifteen most intense precursor ions from each full scan (resolution 60,000) were isolated for HCD MS2 (resolution 15,000; NCE 27) with a dynamic exclusion time of 30 s. Precursors with 1+, 2+, more than 7+ or unassigned charge states were excluded.

## Identification of cross-linked peptides using pLink 2

The pLink 2 (*Chen et al., 2019*) search parameters were as follows: instrument, HCD; precursor mass tolerance, 20 ppm; fragment mass tolerance 20 ppm; cross-linker, BS3/DSS (cross-linking sites K and protein N terminus, cross-link mass-shift 138.068, mono-link mass-shift 156.079); peptide length, minimum six amino acids and maximum 60 amino acids per chain; peptide mass, minimum 600 and maximum 6000 Da per chain; enzyme, Trypsin, with up to three missed cleavage sites per chain; Carbamidomethyl[C], Oxidation[M], Deamidated[N] and Deamidated[Q] as variable modification. The results were filtered by requiring FDR < 1%, E-value <0.0001, spectra count >2.

## Differential Scanning Fluorimetry (DSF)

CDK12KD/CCNKΔC (WT or G731E/R) and DDB1ΔBPB were diluted to 5 μM with DSF assay buffer (50 mM HEPES pH 7.5, 300 mM NaCl, 0.5 mM TCEP). 50 μM HQ461 or 10 μM HBx was added to diluted proteins (final DMSO concentration 0.25%). Twenty μL of protein-compound mix was aliquoted per qPCR tube (Labtidebiotech, Shanghai, China, P01-0803E) followed by the addition of 5 μL of 1:400 diluted SYPRO Orange (Sigma-Aldrich, MO, USA, S5692). After incubation on ice for 20 min, fluorescence measurements were performed using a Bio-Rad CFX96 Realtime PCR instrument. During the DSF experiment, the temperature was increased from 10°C to 95°C at an increment of 0.5 °C with equilibration time of 10 s at each temperature prior to measurement. Tm was defined as the temperature corresponding to the maximum value of the first derivative of fluorescence transition.

## Nano DSF

CDK12KD/CCNKΔC (WT or G731E/R) or DDB1ΔBPB were diluted with DSF assay buffer (50 mM HEPES pH 7.5, 300 mM NaCl, 0.5 mM TCEP) to 3.6 μM and mixed with diluted HQ461 (final DMSO concentration 2%). Tryptophan fluorescence was measured using Tycho NT.6 (NanoTemper, South San Francisco, USA). During the thermal shift experiment, temperature was increased from 35°C to 95°C and fluorescence was scanned at 350 nm and 330 nm. Ti (inflection temperature of the unfolding transition) was calculated from the 350 nm/330 nm ratio.

## CCNK-*luc* degradation assay

The ORF for CCNK was fused with the firefly luciferase (*luc*) sequence and inserted into the pCDNA3.1 vector. For setting up cells for co-transfection, 0.6 million 293 T cells were seeded per well in 6-well plates and allowed to attach to plates overnight. These cells were then transfected with 50 ng of pCDNA3.1-3xFLAG-CDK12 kinase domain, 50 ng of pCDNA3.1-CCNK-luc, and 400 ng of the empty pCDNA3.1 vector. Twenty-four hours later, transfected cells were dissociated by trypsin digestion and then plated in 96-well white plates. After overnight attachments, cells were then treated with HQ461 and its analog derivatives for 8 hr followed by measurements of luciferase activity with Bright-Glo luciferase assay system (Promega, Madison, USA). Half maximal CCNK-degradation concentrations ($DC_{50}$) of HQ461 and its active analogs were interpolated by curve fitting with GraphPad Prism using baseline correction (by normalizing to DMSO control), the asymmetric (five parameter) equation, and least squares fit.

## RNA isolation and RT-qPCR

Total RNA was extracted from HQ461-treated A549 cells using TRNzol (Tiangen, Beijing, China). Reverse transcription of mRNA into cDNA was performed using the 5X All-In-One RT mastermix kit with AccuRT genomic DNA removal kit (Abm, Richmond, Canada), and qPCR was performed using TB Green Premix Ex Taq (Tli RNaseH Plus) (Takara, Otsu, Japan) with the CFX96 Real-Time PCR Detection System (Bio-Rad, Hercules, USA). GAPDH was used for normalization. Primer sequences are as follows: GAPDH-forward: 5'-GCTCTCTGCTCCTCCTGTTC-3'; GAPDH-reverse: 5'-ACGAC-CAAATCCGTTGACTC-3'; BRCA1-forward: 5'-GGCTGTGGGGTTTCTCAGAT-3'; BRCA1-reverse: 5'-TTCATGGAGCAGAACTGGTG-3'; BRCA2-forward: 5'-TTCATGGAGCAGAACTGGTG-3'; BRCA2-reverse: 5'-AGGAAAAGGTCTAGGGTCAGG-3'; ATR-forward: 5'-CGCTGAACTGTACGTGGAAA-3'; ATR-reverse: 5'-CAATAAGTGCCTGGTGAACATC-3'; ERCC4-forward: 5'-TAGACCTAGTAAGAGG-CACAG-3'; ERCC4- reverse: 5'-GAGTGAGGATGTAATCTCCAA-3'. Each experiment was performed with three technical replicates.

## Colony formation assay after CCNK depletion by CRISPR-Cas9

An sgRNA sequence (5'-GTGAACCGAGCGCCCTCTCGG-3') targeting *CCNK* was cloned into lenti-CRISPR v2. An sgRNA-resistant ORF of *CCNK* (by replacing the sgRNA target sequence to 5'-taTcgTcgTgaAggTgcGcgCttTa-3') was cloned into lentiCas9-blast by replacing Cas9 ORF with the sgRNA-resistant *CCNK* ORF. The resulting plasmid was packaged into lentiviral vector to stably infect A549 cells. Parental A549 cells or A549 cells expressing sgRNA-resistant *CCNK* were infected with high MOI with lentivirus carrying sg*NTC* or sg*CCNK*. One day post infection, 1000 cells were plated per well in a six-well tissue-culture plate. Cells were grown for 10 days with medium change every 2 to 3 days. The resulting cell colonies were fixed in 4% PFA in PBS, stained with crystal violet (Beyotime, Shanghai, China, C0121) for 20 min at room temperature, and then de-stained with water. Images were taken with VILBER FX7 imager.

## Preparation of analogues of HQ461

All reactions were carried out under an atmosphere of nitrogen in flame-dried glassware with magnetic stirring unless otherwise indicated. Commercially obtained reagents were used as received. Solvents were dried by passage through an activated alumina column under argon. Liquids and solutions were transferred via syringe. All reactions were monitored by thin-layer chromatography with E. Merck silica gel 60 F254 pre-coated plates (0.25 mm). Structures of the target compounds in this work were assigned by use of NMR spectroscopy and MS spectrometry. The purities of all compounds were >95% as determined on Waters HPLC with 2998PDA and 3100 MS detectors, using ESI as ionization. Pre-HPLC is used to separate and refine high-purity target compounds. $^1$H and $^{13}$C NMR spectra were recorded on Varian Inova-400 spectrometers. Data for $^1$H NMR spectra are reported relative to $CDCl_3$ (7.26 ppm), $CD_3OD$ (3.31 ppm), or DMSO-$d_6$ (2.50 ppm) as an internal standard and are reported as follows: chemical shift (δ ppm), multiplicity (s = singlet, d = doublet, t = triplet, q = quartet, sept = septet, m = multiplet, br = broad), coupling constant *J* (Hz), and integration.

## Acknowledgements

We thank Steven McKnight, Xiaodong Wang, and Deepak Nijhawan for helpful discussions, Deepak Nijhawan, Linfeng Sun, Sanduo Zheng, Feiran Lu, and Dapeng Ju for cell lines and reagents, and John Snyder for editing our manuscript. This work was supported by institutional grants from the Chinese Ministry of Science and Technology, Beijing Municipal Commission of Science and Technology, and Tsinghua University.

## Additional information

### Competing interests

Lu Lv, Peihao Chen, Longzhi Cao, Yamei Li, Qingcui Wu, Jiaojiao Li, Ting Han: A provisional patent application (PCT/CN2020/095482) has been filed for the application of HQ461 and related small molecules as molecular glues regulating CDK12-DDB1 interaction to degrade CCNK. The other authors declare that no competing interests exist.

### Funding

| Funder | Author |
| --- | --- |
| Chinese Ministry of Science and Technology | Xiangbing Qi Ting Han |
| Beijing Municipal Commission of Science and Technology | Xiangbing Qi Ting Han |
| Tsinghua University | Xiangbing Qi Ting Han |

The funders had no role in study design, data collection and interpretation, or the decision to submit the work for publication.

### Author contributions

Lu Lv, Data curation, Formal analysis, Validation, Investigation, Visualization, Methodology, Writing - original draft; Peihao Chen, Longzhi Cao, Data curation, Formal analysis, Investigation, Methodology, Writing - original draft; Yamei Li, Data curation, Formal analysis, Validation, Investigation, Methodology; Zhi Zeng, Data curation, Software, Formal analysis, Visualization, Methodology, Writing - original draft; Yue Cui, Data curation, Investigation, Methodology; Qingcui Wu, Jiaojiao Li, Resources, Project administration; Jian-Hua Wang, Resources, Data curation, Formal analysis, Methodology; Meng-Qiu Dong, Resources, Methodology; Xiangbing Qi, Conceptualization, Resources, Formal analysis, Supervision, Funding acquisition, Methodology, Writing - original draft, Project administration, Writing - review and editing; Ting Han, Conceptualization, Resources, Data curation, Formal analysis, Supervision, Funding acquisition, Investigation, Methodology, Writing - original draft, Project administration, Writing - review and editing

### Author ORCIDs

Lu Lv https://orcid.org/0000-0003-2075-8620
Meng-Qiu Dong http://orcid.org/0000-0002-6094-1182
Xiangbing Qi https://orcid.org/0000-0002-7139-5164
Ting Han https://orcid.org/0000-0003-3168-8699

### Decision letter and Author response

Decision letter https://doi.org/10.7554/eLife.59994.sa1
Author response https://doi.org/10.7554/eLife.59994.sa2

## Additional files

### Supplementary files

• Transparent reporting form

### Data availability

All data generated or analysed during this study are included in the manuscript and supporting files. Sequencing data have been deposited in GEO (GSE153700 and GSE153707).

The following datasets were generated:

| Author(s) | Year | Dataset title | Dataset URL | Database and Identifier |
|---|---|---|---|---|
| Lv L, Chen P, Cao L, Li Y, Zeng Z, Cui Y, Wu Q, Li J, Wang JH, Dong MQ, Qi X, Han T | 2020 | Discovery of a molecular glue promoting CDK12-DDB1 interaction to trigger Cyclin K degradation [CRISPR] | https://www.ncbi.nlm.nih.gov/geo/query/acc.cgi?acc=GSE153700 | NCBI Gene Expression Omnibus, GSE153700 |
| Lv L, Chen P, Cao L, Li Y, Zeng Z, Cui Y, Wu Q, Li J, Wang JH, Dong MQ, Qi X, Han T | 2020 | Discovery of a molecular glue promoting CDK12-DDB1 interaction to trigger Cyclin K degradation [WES] | https://www.ncbi.nlm.nih.gov/geo/query/acc.cgi?acc=GSE153707 | NCBI Gene Expression Omnibus, GSE153707 |

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

## Appendix 1

## General procedure A

Reagents and conditions: (a) 2, Xylene, 135°C, 84%; (b) Br$_2$, AcOH, 60°C, 10min, 73%; (c) 5, EtOH, reflux, 12h.

**Appendix 1—scheme 1.** General procedure A.

**Appendix 1—chemical structure 1.** Synthesis of N-(5-methylthiazol-2-yl)−3-oxobutanamide (3). To a solution of 5-methylthiazol-2-amine (1) (0.5 g; 4.39 mmol) in xylene (10 ml) was added 2, 2-dimethyl-4H-1, 3-dioxin-4-one (2) (685 mg; 4.83 mmol). The mixture was stirred for 12 hr at 135°C, then cooled to room temperature. The solution was filtered and washed with petroleum ether, after filtration, the solid was used directly without purification (3) (0.73 g; 3.69 mmol, 84%). $^1$H NMR (400 MHz, DMSO-$d_6$) δ 11.95 (s, 1H), 7.13 (s, 1H), 3.66 (s, 2H), 2.34 (s, 3H), 2.19 (s, 3H); $^{13}$C NMR (101 MHz, DMSO-$d_6$) δ 202.68, 165.26, 156.30, 135.18, 126.70, 51.22, 30.70, 11.53; HRMS (*m/z*): [M+H]$^+$ calculated for C$_8$H$_{11}$N$_2$O$_2$S, 199.0530; found, 199.0548.

**Appendix 1—chemical structure 2.** Synthesis of 4-bromo-N-(5-methylthiazol-2-yl)−3-oxobutana-mide (4). To a solution of N-(5-methylthiazol-2-yl)−3-oxobutanamide (3) (2.0 g; 10.1 mmol) in AcOH (30 ml) was added Br$_2$ (2.0 g; 11.1 mmol) at 60°C. The mixture was stirred for 10 min at 60°C, then cooled to room temperature. The reaction mixture was quenched with saturated aqueous Na$_2$SO$_3$ solution, extracted with EtOAc and the organic layer dried over Na$_2$SO$_4$, filtered and evaporated. The product was purified by column chromatography on silica gel to obtain 4-bromo-N-(5-methylthiazol-2-yl)−3-oxobutan amide (4) (2.0 g; 7.25 mmol, 73%). $^1$H NMR (400 MHz, DMSO-$d_6$) δ 12.02 (s, 1H), 7.14 (s, 1H), 4.48 (s, 2H), 3.83 (s, 2H), 2.34 (s, 3H). $^{13}$C NMR (101 MHz, DMSO-$d_6$) δ 196.04, 164.67, 156.24, 135.15, 126.89, 47.91, 37.59, 11.55; HRMS (*m/z*): [M+H]$^+$ calculated for C$_8$H$_{10}$BrN$_2$O$_2$S, 276.9641; found, 276.9659.

**Appendix 1—chemical structure 3.** Synthesis of the HQ461 and analogues-A. To a solution of above 4 (10 mg; 0.036 mmol) in EtOH (2 ml) was added the pyridinyl thiourea 5 (0.054 mmol). The mixture was stirred for 12 hr at 70°C, then cooled to room temperature. The mixture was evaporated to dryness and purified by silica gel chromatography to get HQ461 analogues as HBr salt.

**HQ461**

**Appendix 1—chemical structure 4.** 2-(2-((6-methylpyridin-2-yl)amino)thiazol-4-yl)-N-(5-methylthia-zol-2-yl)acetamide (HQ461). Pale yellow solid; Yield 52%; $^1$H NMR (400 MHz, DMSO-$d_6$) δ 12.12 (s, 1H), 11.67 (s, 1H), 7.66 (t, *J* = 7.7 Hz, 1H), 7.14 (d, *J* = 1.2 Hz, 1H), 6.88 (t, *J* = 10.2 Hz, 3H), 3.85 (s,

2H), 2.47 (s, 3H), 2.33 (s, 3H); $^{13}$C NMR (101 MHz, DMSO-$d_6$) δ 167.50, 161.13, 156.59, 154.46, 149.70, 140.66, 138.69, 134.87, 126.86, 117.62, 110.42, 109.70, 36.39, 22.99, 11.61; HRMS (m/z): [M+H]$^+$ calculated for $C_{15}H_{16}N_5OS_2$, 346.0791; found, 346.0847.

**Appendix 1—chemical structure 5.** 2-(2-((6-fluoropyridin-2-yl)amino)thiazol-4-yl)-N-(5-methylthiazol-2-yl)acetamide (HQ001). Pale yellow solid; Yield 50%; $^1$H NMR (400 MHz, CDCl$_3$: CD$_3$OD = 4:1) δ 7.53 (dd, J = 15.9, 7.9 Hz, 1H), 6.87 (s, 1H), 6.60 (d, J = 7.7 Hz, 1H), 6.49 (s, 1H), 6.28 (d, J = 7.8 Hz, 1H), 3.61 (s, 2H), 2.20 (s, 3H). $^{19}$F NMR (376 MHz, CDCl$_3$: CD$_3$OD = 4:1) δ −66.70; HRMS (m/z): [M+H]$^+$ calculated for $C_{14}H_{13}FN_5OS_2$, 350.0540; found, 350.0564.

**Appendix 1—chemical structure 6.** 2-(2-((6-chloropyridin-2-yl)amino)thiazol-4-yl)-N-(5-methylthiazol-2-yl)acetamide (HQ002). Pale yellow solid; Yield 35%; $^1$H NMR (400 MHz, CDCl$_3$: CD$_3$OD = 4:1) δ 7.52 (dt, J = 11.0, 7.6 Hz, 1H), 7.01 (s, 1H), 6.89–6.82 (t, J = 7.2 Hz, 1H), 6.78 (dd, J = 8.2, 3.2 Hz, 1H), 6.58 (s, 1H), 3.76 (s, 2H), 2.35 (s, 3H). HRMS (m/z): [M+H]$^+$ calculated for $C_{14}H_{13}ClN_5OS_2$, 366.0245; found, 366.0265.

**Appendix 1—chemical structure 7.** 2-(2-((6-bromopyridin-2-yl)amino)thiazol-4-yl)-N-(5-methylthiazol-2-yl)acetamide (HQ003). Pale yellow solid; Yield 53%; $^1$H NMR (400 MHz, CDCl$_3$: CD$_3$OD = 4:1) δ 7.41 (t, J = 7.9 Hz, 1H), 6.99 (m, 2H), 6.79 (d, J = 8.3 Hz, 1H), 6.58 (s, 1H), 3.74 (s, 2H), 2.32 (s, 3H); HRMS (m/z): [M+H]$^+$ calculated for $C_{14}H_{13}BrN_5OS_2$, 409.9739; found, 409.9784.

**Appendix 1—chemical structure 8.** 2-(2-((6-ethylpyridin-2-yl)amino)thiazol-4-yl)-N-(5-methylthiazol-2-yl)acetamide (HQ004). Pale yellow solid; Yield 53%; $^1$H NMR (400 MHz, CDCl$_3$: CD$_3$OD = 4:1) δ 7.52 (t, J = 7.8 Hz, 1H), 7.06 (s, 1H), 6.76 (d, J = 7.6 Hz, 1H), 6.64 (d, J = 7.8 Hz, 1H), 6.59 (s, 1H), 3.79 (s, 2H), 2.83 (q, J = 7.2, 14.8 Hz, 2H), 2.39 (s, 3H), 1.38 (t, J = 7.2, 3H); HRMS (m/z): [M+H]$^+$ calculated for $C_{16}H_{18}N_5OS_2$, 360.0947; found, 360.0967.

**Appendix 1—chemical structure 9.** 2-(2-((6-(hydroxymethyl)pyridin-2-yl)amino)thiazol-4-yl)-N-(5-methylthiazol-2-yl)acetamide (HQ005). Pale yellow solid; Yield 33%; $^1$H NMR (400 MHz, CDCl$_3$) δ 11.93 (s, 1H), 9.83 (s, 1H), 7.61 (t, J = 7.8 Hz, 1H), 7.08 (s, 1H), 6.86 (d, J = 7.6 Hz, 1H), 6.72 (d, J = 7.7 Hz, 1H), 6.63 (s, 1H), 4.80 (s, 2H), 3.80 (s, 2H), 2.40 (s, 3H); HRMS (m/z): [M+H]$^+$ calculated for $C_{15}H_{16}N_5O_2S_2$, 362.0740; found, 362.0768.

**Appendix 1—chemical structure 10.** Tert-butyl(6-((4-(2-((5-methylthiazol-2-yl)amino)−2-oxoethyl)thiazol-2-yl)amino)pyridin-2-yl)carbamate (HQ007-01). White solid; Yield 35%; $^1$H NMR (400 MHz, CDCl$_3$: CD$_3$OD = 4:1) δ 7.60 (t, $J$ = 7.6 Hz, 1H), 7.50 (d, $J$ = 7.7 Hz, 1H), 7.06 (s, 1H), 6.56 (s, 1H), 6.51 (d, $J$ = 7.9 Hz, 1H), 3.79 (s, 2H), 2.39 (s, 3H), 1.549 (s, 9H); HRMS ($m/z$): [M+H]$^+$ calculated for C$_{19}$H$_{23}$N$_6$O$_3$S$_2$, 447.1268; found, 447.1295.

**HQ007**

**Appendix 1—chemical structure 11.** 2-(2-((6-aminopyridin-2-yl)amino)thiazol-4-yl)-N-(5-methylthiazol-2-yl)acetamide (HQ007). To a solution of HQ007-01 (3 mg) in dichloromethane (1 ml) was added trifluoroacetic acid (0.5 ml). The mixture was stirred for 2 hr at room temperature. The mixture was evaporated to dryness and purified by silica gel chromatography to get 2-(2-((6-aminopyridin-2-yl)amino)thiazol-4-yl)-N-(5-methylthiazol-2-yl)acetamide as TFA salt (2.4 mg, 80%). $^1$H NMR (400 MHz, CDCl$_3$: CD$_3$OD = 4:1) δ 7.42 (t, $J$ = 8.3 Hz, 1H), 6.83 (s, 1H), 6.66 (s, 1H), 6.13 (d, $J$ = 7.9 Hz, 1H), 6.06 (d, $J$ = 8.1 Hz, 1H), 3.65 (s, 2H), 2.13 (s, 3H); HRMS ($m/z$): [M+H]$^+$ calculated for C$_{14}$H$_{15}$N$_6$OS$_2$, 347.0743; found, 347.0777.

**HQ009**

**Appendix 1—chemical structure 12.** N-(5-methylthiazol-2-yl)−2-(2-((6-(trifluoromethyl)pyridin-2-yl)amino)thiazol-4-yl)acetamide (HQ009). White solid; Yield 52%; $^1$H NMR (400 MHz, CDCl$_3$: CD$_3$OD = 4:1) δ 7.71 (t, $J$ = 8.1 Hz, 1H), 7.18 (d, $J$ = 7.6 Hz, 1H), 7.03 (d, $J$ = 8.1 Hz, 1H), 6.99 (s, 1H), 6.59 (s, 1H), 3.75 (s, 2H), 2.33 (s, 3H). $^{19}$F NMR (376 MHz, CDCl$_3$: CD$_3$OD = 4:1) δ −68.59; HRMS ($m/z$): [M+Na]$^+$ calculated for C$_{15}$H$_{12}$F$_3$N$_5$OS$_2$Na, 422.0333; found, 422.0359.

**HQ010**

**Appendix 1—chemical structure 13.** N-(5-methylthiazol-2-yl)−2-(2-(pyridin-2-ylamino)thiazol-4-yl)acetamide (HQ010). Pale yellow solid; Yield 53%; $^1$H NMR (400 MHz, CDCl$_3$: CD$_3$OD = 4:1) δ 8.28 (d, $J$ = 4.6 Hz, 1H), 7.57 (t, $J$ = 7.4 Hz, 1H), 6.99 (s, 1H), 6.85 (m, 2H), 6.51 (s, 1H), 3.73 (s, 2H), 2.33 (s, 3H); HRMS ($m/z$): [M+H]$^+$ calculated for C$_{14}$H$_{14}$N$_5$OS$_2$, 332.0634; found, 332.0663.

## Appendix 2

## General procedure B

Reagents and conditions: (a) 7, EDCI, DMAP, DMF, rt, 3h; (b) 9, Pd$_2$(dba)$_3$, t-BuOK, Xantphos, 1,4-dioxane, 110℃, 12h.

**Appendix 2—scheme 1.** General procedure B.

HQ011

**Appendix 2—chemical structure 1.** Synthesis of 2-(2-aminothiazol-4-yl)-N-(5-methylthiazol-2-yl)acet-amide (HQ011). To a mixture of 2-(2-aminothiazol-4-yl)acetic acid (6) (1.00 g; 6.33 mmol) and 5-methylthiazol-2-amine (1) (722 mg; 6.33 mmol) in dimethylformamide (15 ml) was added 1-Ethyl-3-(3-dimethylaminopropyl) carbodiimide hydrochloride (1.82 g; 9.50 mmol) and dimethylaminopyridine (154 mg; 1.27 mmol) under argon atmosphere. The mixture stirred at room temperature for 3 hr. The reaction mixture was quenched with water, extracted with EtOAc and the organic layer dried over Na$_2$SO$_4$, filtered and evaporated. The product was purified by column chromatography on silica gel to obtain 2-(2-aminothiazol-4-yl)-N-(5-methylthiazol-2-yl) acetamide (**7**) (1.15 g; 4.53 mmol, 72%). $^1$H NMR (400 MHz, DMSO-$d_6$) δ 11.97 (s, 1H), 7.12 (s,, 1H), 6.95 (s, 2H), 6.32 (s, 1H), 3.56 (s, 2H), 2.33 (s, 3H); $^{13}$C NMR (101 MHz, DMSO-$d_6$) δ 168.82, 168.27, 156.59, 145.33, 135.17, 126.54, 103.53, 38.52, 11.57; HRMS (*m/z*): [M+Na]$^+$ calculated for C$_9$H$_{10}$N$_4$OS$_2$Na, 277.0194; found, 277.0209.

HQ013

**Appendix 2—chemical structure 2.** 2-(2-aminothiazol-4-yl)-N-(4-(furan-2-yl)thiazol-2-yl)acetamide (HQ013). To a mixture of 2-(2-aminothiazol-4-yl)acetic acid (6) (100 mg; 0.63 mmol) and 4-(furan-2-yl) thiazol-2-amine (105 mg; 0.63 mmol) in dimethylformamide (5 ml) was added 1-Ethyl-3-(3-dimethylaminopropyl) carbodiimide hydrochloride (182 mg; 0.95 mmol) and dimethylaminopyridine (15 mg; 0.13 mmol) under argon atmosphere. The mixture stirred at room temperature for 3 hr. The reaction mixture was quenched with water, extracted with EtOAc and the organic layer dried over Na$_2$SO$_4$, filtered and evaporated. The product was purified by column chromatography on silica gel to obtain 2-(2-aminothiazol-4-yl)-N-(4-(furan-2-yl)thiazol-2-yl)acetamide (HQ013) (124 mg; 0.41 mmol, 65%). $^1$H NMR (400 MHz, DMSO-$d_6$) δ 12.43 (s, 1H), 7.72 (dd, *J* = 1.8, 0.8 Hz, 1H), 7.32 (s, 1H), 6.93 (s, 2H), 6.67 (d, *J* = 2.7 Hz, 1H), 6.58 (dd, *J* = 3.3, 1.8 Hz, 1H), 6.35 (s, 1H), 3.60 (s, 2H); HRMS (*m/z*): [M+H]$^+$ calculated for C$_{12}$H$_{11}$N$_4$O$_2$S$_2$, 307.0318; found, 307.0337.

**Appendix 2—chemical structure 3.** Synthesis of the HQ461 and analogues-B. A mixture of HQ011 (20 mg, 0.08 mmol), 9 (0.096 mmol), Xantphos (5.6 mg, 0.0096 mmol) Tris(dibenzylideneacetone) dipalladium (8.8 mg, 0.0096 mmol), tBuOK (21 mg, 0.19 mmol) in 1,4-dioxane(2.0 ml) under argon atmosphere was stirred at 110℃ for 12 hr. The mixture was filtered and concentrated in vacuum and the crude product purified by column chromatography on silica gel.

**HQ006**

**Appendix 2—chemical structure 4.** 2-(2-((6-methoxypyridin-2-yl)amino)thiazol-4-yl)-N-(5-methylthia-zol-2-yl)acetamide (HQ006). Pale yellow solid; Yield 33%; $^1$H NMR (400 MHz, CDCl$_3$) δ 11.60 (s, 1H), 9.36 (s, 1H), 7.51 (t, $J$ = 7.9 Hz, 1H), 7.09 (s, 1H), 6.62 (s, 1H), 6.35 (dd, $J$ = 7.9, 3.3 Hz, 2H), 4.08 (s, 3H), 3.79 (s, 2H), 2.40 (s, 3H); HRMS (m/z): [M+H]$^+$ calculated for C$_{15}$H$_{16}$N$_5$O$_2$S$_2$, 362.0740; found, 362.0768.

**HQ008**

**Appendix 2—chemical structure 5.** 2-(2-((6-(1H-pyrazol-1-yl)pyridin-2-yl)amino)thiazol-4-yl)-N-(5-methylthiazol-2-yl)acetamide (HQ008). Pale yellow solid; Yield 25%; $^1$H NMR (400 MHz, DMSO-$d_6$) δ 12.09 (s, 1H), 11.63 (s, 1H), 8.75 (d, J = 2.2 Hz, 1H), 7.91–7.81 (m, 2H), 7.42 (d, J = 7.9 Hz, 1H), 7.14 (s, 1H), 6.94–6.88 (m, 2H), 6.64 (s, 1H), 3.80 (s, 2H), 2.33 (s, 3H); HRMS (m/z): [M+H]$^+$ calculated for C$_{17}$H$_{16}$N$_7$OS$_2$, 398.0852; found, 398.0882.

**HQ012**

**Appendix 2—chemical structure 6.** N-(4-(furan-2-yl)thiazol-2-yl)−2-(2-((6-methylpyridin-2-yl)amino) thiazol-4-yl)acetamide (HQ012). A mixture of HQ013 (20 mg, 0.08 mmol), 2-bromo-6-methylpyridine (16.4 mg, 0.096 mmol), Xantphos (5.6 mg, 0.0096 mmol), Tris(dibenzylideneacetone)dipalladium (8.8 mg, 0.0096 mmol), t-BuOK (21 mg, 0.19 mmol) in 1,4-dioxane (2.0 ml) under argon atmosphere was stirred at 110°C for 12 hr. The mixture was filtered and concentrated in vacuum and the crude product purified by column chromatography on silica gel to obtain N-(4-(furan-2-yl)thiazol-2-yl)−2-(2-((6-methylpyridin-2-yl)amino)thiazol-4-yl) acetamide (HQ012) (9.8 mg; 0.026 mmol, 21%). $^1$H NMR (400 MHz, CDCl$_3$) δ 11.99 (s, 1H), 9.40 (s, 1H), 7.54–7.47 (m, 1H), 7.43 (d, J = 1.8 Hz, 1H), 7.08 (s, 1H), 6.76 (d, J = 7.4 Hz, 1H), 6.69 (d, J = 3.3 Hz, 1H), 6.64 (d, J = 8.1 Hz, 1H), 6.57 (s, 1H), 6.48–6.43 (m, 1H), 3.82 (s, 2H), 2.55 (d, J = 2.9 Hz, 3H); HRMS (m/z): [M+H]$^+$ calculated for C$_{18}$H$_{16}$N$_5$O$_2$S$_2$, 398.0740; found, 398.0784.

**HQ014**

**Appendix 2—chemical structure 7.** Synthesis of 6-bromo-N-(4-(2-((5-methylthiazol-2-yl)amino)−2-oxoethyl)thiazol-2-yl) Picolinamide (HQ014). A mixture of HQ011 (25 mg, 0.10 mmol), 6-bromopicolinic acid (20 mg, 0.10 mmol), 1-Ethyl-3-(3-dimethylaminopropyl) carbodiimide hydrochloride (28 mg, 0.15 mmol) dimethylaminopyridine (1.22 mg, 0.01 mmol) in dimethylformamide (2.0 ml) under argon atmosphere was stirred at room temperature for 12 hr. The reaction mixture was quenched with water, extracted with EtOAc and the organic layer dried over Na$_2$SO$_4$, filtered and evaporated. The product was purified by column chromatography on silica gel to obtain HQ014 (28 mg; 0.064 mmol, 83%). $^1$H NMR (400 MHz, CDCl$_3$: CD$_3$OD = 4:1) δ 8.18 (dd, J = 7.5, 0.6 Hz, 1H), 7.76 (t, J = 7.7 Hz, 1H), 7.71–7.66 (m, 1H), 6.99 (s, 1H), 6.87 (s, 1H), 3.82 (s, 2H), 2.32 (s, 3H); HRMS (m/z): [M+Na]$^+$ calculated for C$_{15}$H$_{12}$BrN$_5$O$_2$S$_2$Na, 459.9513; found, 459.9541.

