## [Decision Letter]

**Acceptance summary:**

This paper describes the identification of a molecular glue (HQ461; HQ for short) that promotes degradation of cyclin K, the activator subunit of CDK12. The authors show biochemically that HQ can promote the recruitment of CDK12 to DDB1-CUL4-RBX1, the core subunits of the CUL4 ubiquitin ligase. Degradation of cyclin K leads to reduced CDK12 (and CDK13) activity and reduced CDK12/13 substrate phosphorylation. Med-chem efforts led to a somewhat more active molecule glue. This work suggests that the appropriate small molecule can bypass a requirement for a substrate-specific adaptor that normally is the target for known molecular glues.

**Decision letter after peer review:**

Thank you for submitting your article "Discovery of a molecular glue promoting CDK12-DDB1 interaction to trigger Cyclin K degradation" for consideration by *eLife*. Your article has been reviewed by three peer reviewers, one of whom is a member of our Board of Reviewing Editors, and the evaluation has been overseen by a Reviewing Editor and Michael Marletta as the Senior Editor. The reviewers have opted to remain anonymous.

The reviewers have discussed the reviews with one another and the Reviewing Editor has drafted this decision to help you prepare a revised submission.

Summary:

This paper describes the identification of a molecular glue (HQ461; HQ for short) that promotes degradation of cyclin K, the activator subunit of CDK12. The authors show biochemically that HQ can promote the recruitment of CDK12 to DDB1-CUL4-RBX1, the core subunits of the CUL4 ubiquitin ligase. Degradation of cyclin K leads to reduced CDK12 (and CDK13) activity and reduced CDK12/13 substrate phosphorylation. Med-chem efforts led to a somewhat more active molecule glue. This work suggests that the appropriate small molecule can bypass a requirement for a substrate-specific adaptor that normally is the target for known molecular glues. Recently, another paper from the Ebert lab identified a different molecule but with a similar activity.

Overall, the reviewers were quite positive about the paper but suggested a couple of aspects that could strengthen the paper, and to a large degree are required to confirm the proposed mechanism.

To facilitate revision, I have left all of the comments of the reviewers below, in addition to the specific requested revisions below.

Essential revisions:

1) Clarify effects on CDK12 in Figure 3, including the TUBE experiment. The reviewers are concerned about the conclusion that HQ promotes degradation of CDK12 in addition to cyclinK. Given that degradation of cyclinK would yield "free" CDK12, it could easily be the case that CDK12 degradation in cells is occurring through a separate mechanism, for example involving an "orphan protein" quality control E3. One approach would be to deplete cyclin K by RNAi and ask whether the half-life of CDK12 is affected and also whether HQ affects the turnover of CDK12. The reviewers also realize that formally distinguish between the possibility that degradation of cyclin K yields "free" CDK12 and the other that CDK12 remains glued by HQ461 may not be trivial. But one approach would be to ask whether CDK12 binds DDB1 in the presence of HQ461 but in the absence of cyclin K? And, if it does, does HQ461 promotes CDK12 ubiquitylation in vitro and degradation in cells?

In terms of in vivo CDK12 ubiquitylation, one idea would be to use TUBEs to ask whether HQ affects CDK12 ubiquitylation in cells.

2) Use CDK12 mutants in the experiments in Figure 4.

3) Redo experiment in Figure 4—figure supplement 1, to correct for loading and also blot for CDK12 ubiquitylation.

4) Cite Ebert paper and maybe discuss differences.

Reviewer #1:

This paper describes the identification of a molecular glue (HQ461; HQ for short) that promotes degradation of cyclin K, the activator subunit of CDK12. The authors show biochemically that HQ can promote the recruitment of CDK12 to DDB1-CUL4-RBX1, the core subunits of the CUL4 ubiquitin ligase. Degradation of cyclin K leads to reduced CDK12 (and CDK13) activity and reduced CDK12/13 substrate phosphorylation. Med-chem efforts led to a somewhat more active molecule glue. This work suggests that the appropriate small molecule can bypass a requirement for a substrate-specific adaptor that normally is the target for known molecular glues.

Recently, another group reported a distinct molecule that functions in a manner similar to HQ reported here. However, the methods of identification are pretty distinct. In this work, the authors found that A549 cells were sensitive to HQ and used a genome-wide CRISPR strategy to identify genes whose deletion made cells resistant to HQ. This led to the identification of components of the CUL4 complex, but interestingly none of the ~30 known substrate specific receptors for CUL4 were identified. This suggested that perhaps one or more proteins might be degraded by a non-canonical CUL4-dependent mechanism. To examine this, the authors performed an interesting genetic screen, identifying CDK12 as a protein whose point mutation (G731E or R) made cells resistant to HQ.

Overall, the biochemical data are OK, but are superseded in many ways by the previous paper which solved the structure of CDK12 bound to DDB1 via a distinct small molecule. Although the authors identify a point mutation that abolishes the activity of HQ, it looks based on the structure to be a residue whose mutation will block interaction with DDB1 (assuming at HQ brings the molecules together in much the same way as the C8 compound), rather than blocking binding of CDK12 to HQ.

So in the end, the authors have identified a new compound and done a reasonable job of showing it works as a molecular glue, but precisely where the molecule is binding and the interactions that facilitate ubiquitylation are not defines structurally. Based on the molecular crosslink identified between DDB1 and CDK12. It seems likely that the interactions are similar, as the crosslinked lysines are within proximity with each other in the C8 structure.

Reviewer #2:

Lv et al., show that HQ461 mediates the polyubiquitination and proteasomal degradation of the CDK12-binding protein Cyclin K (CCNK) and promotes cytotoxicity by inducing the interaction between CDK12 and the DDB1-CUL4-RBX1 E3 complex. Together these results provide a clear and interesting story about the identification of HQ461 as a novel molecular glue. I only have a few suggestions to strengthen the biochemical evidence.

1) Figure 3B and 3C: HQ461 mediates the degradation of both CDK12 and CCNK at a similar level. Why do the authors claim that HQ461 specifically mediates the degradation of CCNK? And if CDK12 is not ubiquitylated by DDB1-CUL4-RBX1, then by which mechanism is CDK12 degraded upon HQ461 treatment?

2) Figure 3D: the authors should evaluate the ubiquitylation of endogenous CDK12 and CCNK (e.g., using TR-TUBE; PMID: 30850049) to strength the hypothesis that CCNK, but not CDK12, is ubiquitylated upon HQ461 treatment.

3) Figure 4A and Figure 4B: the authors should include the G731E and G731R CDK12 mutants (which do not support CCNK degradation). Are these two mutants unable to bind DDB1 in the presence of HQ461? Can DDB1 bind CDK12 in the absence of CCNK?

4) In Figure 4—figure supplement 1, the expression of CCNK in lane 2 and lane 7 appear much less than that in lane 8. The authors should load equal amount of CCNK to be able to compare its ubiquitylation level.

5) Figure 4: the authors should also evaluate CDK12 ubiquitylation in the same experiment.

6) Known molecular glues have high affinity for the E3 than for the substrate (although, theoretically, it could be the opposite) and the affinity further increases in the presence of the three components. The authors should measure the affinity of HQ461 for CDK12 and DDB1, alone or in combination.

Reviewer #3:

Using phenotype-based high-throughput screening, the authors identified HQ461 to possess potent cytotoxicity, and in searching of the underlying mechanisms, combing chemical genetics and biochemical reconstitution, the authors identified HQ461 as a molecular glue that acts by binding to CDK12' kinase domain to recruit Cul4/DDB1 complex, promoting the degradation of Cyclin K. The authors further defined G731E mutation to confer resistant to HQ461-induced Cyclin K degradation and cytotoxicity. The paper is clearly written however, the following concerns should be addressed before its publication at *ELife*.

1) The authors should mention and discuss the recent paper by Ebert group showing that Cdk inhibitor CR8 can function as a molecular glue to trigger Cyclin K degradation.

2) Figure 2A, is there mutation identified in Cyclin K? Will G731E mutation affect CDK12 kinase activity or its protein stability?

3) Figure 3A-3B, the authors should explain the discrepancy of the results. In Figure 3A, 10uM HQ461 led to 50% reduction of CDK12 while in Figure 3B, more than 80% of CDJK12 is degraded after 10uM HQ461 treatment.

4) Figure 3E, it will be nice to know if the kinase activity of CDK12 is important for HQ461-induced degradation of Cyclin K. It will be nice to include a kinase-dead version of CDK12 in this assay.

5) Figure 4A, it will be important for the authors to include the G731E and G731R mutants in this assay to examine whether the G731E and G731R mutation abolished HQ461-mediated interaction between CDK12 and DDB1.

6) Figure 4B, as in Figure 1A, Cul4B and Rbx1 was screened out to be important factors mediating the cytotoxicity of HQ461, it will be important to show whether Cul4B but not Cul4A can be recruited to CDK12 by HQ461 and depleting of Cul4B, but not Cul4A can confer resistance to HQ461-induced degradation of Cyclin K.

---

## [Author Response]

Summary:This paper describes the identification of a molecular glue (HQ461; HQ for short) that promotes degradation of cyclin K, the activator subunit of CDK12. The authors show biochemically that HQ can promote the recruitment of CDK12 to DDB1-CUL4-RBX1, the core subunits of the CUL4 ubiquitin ligase. Degradation of cyclin K leads to reduced CDK12 (and CDK13) activity and reduced CDK12/13 substrate phosphorylation. Med-chem efforts led to a somewhat more active molecule glue. This work suggests that the appropriate small molecule can bypass a requirement for a substrate-specific adaptor that normally is the target for known molecular glues. Recently, another paper from the Ebert lab identified a different molecule but with a similar activity.Overall, the reviewers were quite positive about the paper but suggested a couple of aspects that could strengthen the paper, and to a large degree are required to confirm the proposed mechanism.

We thank the reviewers for their positive comments and valuable suggestions. We believe we have tried our best to experimentally, or textually, address all of these comments. In addition to our point-by-point responses below, we complied a table to highlight modifications to the figures to facilitate evaluation of our revised manuscript.

Essential revisions:1) Clarify effects on CDK12 in Figure 3, including the TUBE experiment. The reviewers are concerned about the conclusion that HQ promotes degradation of CDK12 in addition to cyclinK. Given that degradation of cyclinK would yield "free" CDK12, it could easily be the case that CDK12 degradation in cells is occurring through a separate mechanism, for example involving an "orphan protein" quality control E3. One approach would be to deplete cyclin K by RNAi and ask whether the half-life of CDK12 is affected and also whether HQ affects the turnover of CDK12. The reviewers also realize that formally distinguish between the possibility that degradation of cyclin K yields "free" CDK12 and the other that CDK12 remains glued by HQ461 may not be trivial. But one approach would be to ask whether CDK12 binds DDB1 in the presence of HQ461 but in the absence of cyclin K? And, if it does, does HQ461 promotes CDK12 ubiquitylation in vitro and degradation in cells?In terms of in vivo CDK12 ubiquitylation, one idea would be to use TUBEs to ask whether HQ affects CDK12 ubiquitylation in cells.

We agree with the reviewers that it is important to clarify the effect of HQ461 on CDK12 in addition to cyclin K. We provide multiple lines of evidence to support the model that CDK12 is not a direct substrate for HQ461-induced polyubiquitination.

First, the apparent discrepancy between Figure 3A and 3B/C are likely a result of batch variations of the polyclonal anti-CDK12 antibody. Figure 3A was prepared with an old batch of CDK12 antibody while Figure 3B and 3C were prepared with a new batch. We thus repeated the experiments in Figure 3A with the new batch of anti-CDK12. By diluting untreated A549 lysates (lanes #1-3 of Figure 3A), we show that a two-fold dilution resulted in an ~80% drop in CDK12 band intensity and now there is no discrepancy among Figure 3A-C. Despite the antibody batch variation issue, we arrive at the same conclusion that CDK12 is depleted by ~50% while CCNK is depleted by greater than 87.5% after 8 hours of treatment with 10 µM HQ461.

Second, we used sgRNA to deplete CCNK, and observed a reduction of CDK12 levels. Although the precise mechanism of CDK12 downregulation is not clear at the moment, our result is consistent with the idea that CDK12 stability is decreased upon loss of CCNK. As the reviewer has pointed out, one possibility is that free CDK12 may be degraded by a quality control E3 and we mentioned this possibility in the discussion. The suggestion to examine whether CDK12 binds DDB1 in the presence of HQ461 but in the absence of cyclin K is technically infeasible because CDK12 is destabilized in the absence of cyclin K in vivo. Likely for the same reason, recombinant expression of CDK12 in insect cells is not feasible without co-expression of cyclin K (PMID: 26597175), preventing us from addressing this question with in vitro reconstitution.

Third, we repeated in vitro ubiquitination assays, blotted for both CCNK and CDK12 (New Figure 4D and Figure 4—figure supplement 1), and clearly demonstrated that CCNK is polyubiquitinated in the presence of HQ461 in vitro while CDK12 is not.

Fourth, we attempted the TUBE experiment suggested by the reviewers (Author response image 1). We were able to use FLAG-TUBE pulldown to efficiently enrich polyubiquitinated proteins from 293T lysates. However, we were not able to observe an enrichment of polyubiquitinated CCNK or CDK12. In contrast, using His-Ub transfected 293T cells treated with a proteasome inhibitor bortezomib, we were able to observe polyubiquitinated CCNK but not CDK12 (Figure 3D and Figure 3—figure supplement 1D). Therefore, we think FLAG-TUBE pulldown may not be as sensitive as His-Ub pulldown in detecting HQ461-induced protein polyubiquitination. We also note that His-Ub pulldown allows the solubilization and purification of polyubiquitinated proteins under denaturing conditions (8M urea), permitting the detection of polyubiquitinated proteins that may be insoluble in non-denaturing conditions.

2) Use CDK12 mutants in the experiments in Figure 4.

We included CDK12 mutants in the experiments in Figure 4 per reviewers’ request (New Figure 4B and Figure 5—figure supplement 1A).

3) Redo experiment in Figure 4—figure supplement 1, to correct for loading and also blot for CDK12 ubiquitylation.

We have repeated in vitro ubiquitination experiments and blotted for CDK12, now loading is equal and lack of in vitro CDK12 polyubiquitination is demonstrated (New Figure 4D and Figure 4—figure supplement 1).

4) Cite Ebert paper and maybe discuss differences.

We cited Ebert and Winter papers and discussed differences of our studies in the discussion.

Reviewer #1:This paper describes the identification of a molecular glue (HQ461; HQ for short) that promotes degradation of cyclin K, the activator subunit of CDK12. The authors show biochemically that HQ can promote the recruitment of CDK12 to DDB1-CUL4-RBX1, the core subunits of the CUL4 ubiquitin ligase. Degradation of cyclin K leads to reduced CDK12 (and CDK13) activity and reduced CDK12/13 substrate phosphorylation. Med-chem efforts led to a somewhat more active molecule glue. This work suggests that the appropriate small molecule can bypass a requirement for a substrate-specific adaptor that normally is the target for known molecular glues.Recently, another group reported a distinct molecule that functions in a manner similar to HQ reported here. However, the methods of identification are pretty distinct. In this work, the authors found that A549 cells were sensitive to HQ and used a genome-wide CRISPR strategy to identify genes whose deletion made cells resistant to HQ. This led to the identification of components of the CUL4 complex, but interestingly none of the ~30 known substrate specific receptors for CUL4 were identified. This suggested that perhaps one or more proteins might be degraded by a non-canonical CUL4-dependent mechanism. To examine this, the authors performed an interesting genetic screen, identifying CDK12 as a protein whose point mutation (G731E or R) made cells resistant to HQ.Overall, the biochemical data are OK, but are superseded in many ways by the previous paper which solved the structure of CDK12 bound to DDB1 via a distinct small molecule. Although the authors identify a point mutation that abolishes the activity of HQ, it looks based on the structure to be a residue whose mutation will block interaction with DDB1 (assuming at HQ brings the molecules together in much the same way as the C8 compound), rather than blocking binding of CDK12 to HQ.So in the end, the authors have identified a new compound and done a reasonable job of showing it works as a molecular glue, but precisely where the molecule is binding and the interactions that facilitate ubiquitylation are not defines structurally. Based on the molecular crosslink identified between DDB1 and CDK12. It seems likely that the interactions are similar, as the crosslinked lysines are within proximity with each other in the C8 structure.

We thank reviewer #1 for the positive comments and informing us the actual number of human DCAFs. We corrected this in the text as follows: “More than 60 human proteins have been proposed as DCAFs (Lee and Zhou, 2007) and fewer than 30 are likely bona fide DCAFs.”

Reviewer #2:Lv et al., show that HQ461 mediates the polyubiquitination and proteasomal degradation of the CDK12-binding protein Cyclin K (CCNK) and promotes cytotoxicity by inducing the interaction between CDK12 and the DDB1-CUL4-RBX1 E3 complex. Together these results provide a clear and interesting story about the identification of HQ461 as a novel molecular glue. I only have a few suggestions to strengthen the biochemical evidence.1) Figure 3B and 3C: HQ461 mediates the degradation of both CDK12 and CCNK at a similar level. Why do the authors claim that HQ461 specifically mediates the degradation of CCNK? And if CDK12 is not ubiquitylated by DDB1-CUL4-RBX1, then by which mechanism is CDK12 degraded upon HQ461 treatment?

We thank reviewer #2 for the positive comments and suggestions. We now provide multiple lines of evidence to support the model that CDK12 is not a direct substrate for HQ461-induced polyubiquitination.

First, the apparent discrepancy between Figure 3A and 3B/C are likely a result of batch variations of the polyclonal anti-CDK12 antibody. Figure 3A was prepared with an old batch of CDK12 antibody while Figure 3B and 3C were prepared with a new batch. We thus repeated the experiments in Figure 3A with the new batch of anti-CDK12. By diluting untreated A549 lysates (lanes #1-3 of Figure 3A), we show that a two-fold dilution resulted in an ~80% drop in CDK12 band intensity and now there is no discrepancy among Figure 3A-C. Despite the antibody batch variation issue, we arrive at the same conclusion that CDK12 is depleted by ~50% while CCNK is depleted by greater than 87.5% after 8 hours of treatment with 10 µM HQ461.

Second, we used sgRNA to deplete CCNK, and observed a reduction of CDK12 levels. Although the precise mechanism of CDK12 downregulation is not clear at the moment, our result is consistent with the idea that CDK12 stability is decreased upon loss of CCNK. One possibility is that free CDK12 may be degraded by a quality control E3 and we discussed this possibility in the text.

Third, we repeated in vitro ubiquitination assays, blotted for both CCNK and CDK12 (New Figure 4D and Figure 4—figure supplement 1), and clearly demonstrated that CCNK is polyubiquitinated in the presence of HQ461 in vitro while CDK12 is not.

2) Figure 3D: the authors should evaluate the ubiquitylation of endogenous CDK12 and CCNK (e.g., using TR-TUBE; PMID: 30850049) to strength the hypothesis that CCNK, but not CDK12, is ubiquitylated upon HQ461 treatment.

We attempted the TUBE experiment suggested by the reviewers (Author response image 1). We were able to use FLAG-TUBE pulldown to efficiently enrich polyubiquitinated proteins from 293T lysates. However, we were not able to observe an enrichment of polyubiquitinated CCNK or CDK12. In contrast, using His-Ub transfected 293T cells treated with a proteasome inhibitor bortezomib, we were able to observe polyubiquitinated CCNK but not CDK12 (Figure 3D and Figure 3—figure supplement 1D). Therefore, we think FLAG-TUBE pulldown may not be as sensitive as His-Ub pulldown in detecting HQ461-induced protein polyubiquitination. We also note that His-Ub pulldown allows the solubilization and purification of polyubiquitinated proteins under denaturing conditions (8M urea), permitting the detection of polyubiquitinated proteins that may be insoluble in non-denaturing conditions.

**Author response image 1. respfig1:** Enrichment of polyubiquitinated proteins by FLAG-TUBE. 293F cells were transiently transfected with plasmid expressing FLAG-TUBE. Twenty-four hours later, transfected cells were treated with 10μM HQ461 for 0, 1, 2, 4 or 8 hours.FLAG-TUBE and its interacting proteins were immunoprecipitated by anti-FLAG antibody.

3) Figure 4A and Figure 4B: the authors should include the G731E and G731R CDK12 mutants (which do not support CCNK degradation). Are these two mutants unable to bind DDB1 in the presence of HQ461? Can DDB1 bind CDK12 in the absence of CCNK?

We have included G731E and G731R CDK12 mutants in these figures. These two mutants are indeed not able to bind DDB1 in the presence of HQ461(New Figure 4B and Figure 5—figure supplement 1A).

The suggestion to examine whether CDK12 binds DDB1 in the presence of HQ461 but in the absence of cyclin K is technically infeasible because CDK12 is destabilized in the absence of cyclin K in vivo. Likely for the same reason, recombinant expression of CDK12 in insect cells is not feasible without co-expression of cyclin K (PMID: 26597175), preventing us from addressing this question with in vitro reconstitution.

4) In Figure 4—figure supplement 1, the expression of CCNK in lane 2 and lane 7 appear much less than that in lane 8. The authors should load equal amount of CCNK to be able to compare its ubiquitylation level.

We have repeated this experiment for equal loading (Figure 4—figure supplement 1).

5) Figure 4: the authors should also evaluate CDK12 ubiquitylation in the same experiment.

We have evaluated CDK12 polyubiquitination in this same experiment (Figure 4—figure supplement 1).

6) Known molecular glues have high affinity for the E3 than for the substrate (although, theoretically, it could be the opposite) and the affinity further increases in the presence of the three components. The authors should measure the affinity of HQ461 for CDK12 and DDB1, alone or in combination.

Using DSF, we qualitatively demonstrate that HQ461 binds CDK12, but not DDB1 (Figure 5C-D). Using Alphascreen assay we show that HQ461 mediates interaction between CDK12 and DDB1 (Figure 5A). Due to restrictions imposed by COVID-19, we were unable to perform additional quantitative biophysical measurements to completely address this comment. We will conduct these experiments and report our findings in the future.

Reviewer #3:Using phenotype-based high-throughput screening, the authors identified HQ461 to possess potent cytotoxicity, and in searching of the underlying mechanisms, combing chemical genetics and biochemical reconstitution, the authors identified HQ461 as a molecular glue that acts by binding to CDK12' kinase domain to recruit Cul4/DDB1 complex, promoting the degradation of Cyclin K. The authors further defined G731E mutation to confer resistant to HQ461-induced Cyclin K degradation and cytotoxicity. The paper is clearly written however, the following concerns should be addressed before its publication at ELife.1) The authors should mention and discuss the recent paper by Ebert group showing that Cdk inhibitor CR8 can function as a molecular glue to trigger Cyclin K degradation.

We cited Ebert and Winter papers and discussed the differences of our studies in the Discussion section.

2) Figure 2A, is there mutation identified in Cyclin K? Will G731E mutation affect CDK12 kinase activity or its protein stability?

We did not identify any mutation in Cyclin K. Using GST-CTD as substrate, we show that G731E does not affect CDK12 kinase activity (Author response image 2). G731E does not affect CDK12 protein stability as measured by DSF (Author response image 2). We prefer to keep these results out of the paper for better readability. Because our responses will be published along with the paper, readers can see these results online.

**Author response image 2. sa2fig2:** CDK12G731E/R mutations do not affect CDK12 kinase activity or protein stability. (A) CDK12KD/CCNKΔC (WT or G731E/R mutants) kinase activity was perform in 10μL reaction volume containing 45nM substrate (GST-CTD) and a reaction buffer of 20mM HEPES, pH7.9, 20mM MgCl2, 2mM DTT, 1mM ATP. The reaction was initiated by adding 8nM, 80nM, 800nM, or 8μM CDK12KD/CCNKΔC (WT or G731E/R mutants) and incubated at 30°C for 1hour followed by the addition of 10μL ADP-GloTM Reagent (Promega, V6930) to terminate kinase reaction and remove unreacted ATP. After incubation at room temperature for 40 minutes, the mixtures were transferred from PCR tubes to a 384-well plate and mixed with 20μL Kinase Detection Reagent (Promega, V6930). The mixtures were allowed to sit at room temperature for 40 minutes before reading luminescence by EnVision multimode plate reader (Perkin Elmer). Error bars represent SD from three replicates. (B) Thermal denaturation of CDK12KD/CCNKΔC (WT or G731E/R mutants) measured by nano DSF.

3) Figure 3A-3B, the authors should explain the discrepancy of the results. In Figure 3A, 10uM HQ461 led to 50% reduction of CDK12 while in Figure 3B, more than 80% of CDJK12 is degraded after 10uM HQ461 treatment.

The apparent discrepancy between Figure 3A and 3B/C are likely a result of batch variations of the polyclonal anti-CDK12 antibody. Figure 3A was prepared with an old batch of CDK12 antibody while Figure 3B and 3C were prepared with a new batch. We thus repeated the experiments in Figure 3A with the new batch of anti-CDK12. By diluting untreated A549 lysates (lanes #1-3 of Figure 3A), we show that a two-fold dilution resulted in an ~80% drop in CDK12 band intensity and now there is no discrepancy among Figure 3A-C. Despite the antibody batch variation issue, we arrive at the same conclusion that CDK12 is depleted by ~50% while CCNK is depleted by greater than 87.5% after 8 hours of treatment with 10 µM HQ461.

4) Figure 3E, it will be nice to know if the kinase activity of CDK12 is important for HQ461-induced degradation of Cyclin K. It will be nice to include a kinase-dead version of CDK12 in this assay.

Using a kinase inactive mutant D877N of CDK12 (PMID:24662513), we show that the kinase activity of CDK12 is not important for HQ461-induced degradation of Cyclin K (Author response image 3). We prefer to keep this result out of the paper for better readability. Because our responses will be published along with the paper, readers can see these results online.

**Author response image 3. sa2fig3:** CDK12 kinase activity is not required for HQ461-induced CCNK degradation.

5) Figure 4A, it will be important for the authors to include the G731E and G731R mutants in this assay to examine whether the G731E and G731R mutation abolished HQ461-mediated interaction between CDK12 and DDB1.

We have included G731E and G731R mutants in this assay and shown the G731E and G731R mutation abolished HQ461-mediated interaction between CDK12 and DDB1 (Figure 4B).

6) Figure 4B, as in Figure 1A, Cul4B and Rbx1 was screened out to be important factors mediating the cytotoxicity of HQ461, it will be important to show whether Cul4B but not Cul4A can be recruited to CDK12 by HQ461 and depleting of Cul4B, but not Cul4A can confer resistance to HQ461-induced degradation of Cyclin K.

CUL4A was also identified in the screen (now labeled on Figure 1C). We used CRISPR to inactivate either CUL4A or CUL4B, both resulted in partial resistance to HQ461 (Author response image 4). This result suggests that CUL4A and CUL4B both participate in HQ461 mediated degradation of cyclin K.

**Author response image 4. sa2fig4:** Measurement of the HQ461 IC50 on the viability of A549 cells expressing non-targeting control (NTC, IC50=2.3 μM) or two independent sgRNAs targeting CUL4A (IC50=4.4 μM, 4.2 μM), or CUL4B (IC50=6.0 μM, 6.3 μM). Error bars represent SEM from three biological replicates.